# Interpreting Any Condition to Caption for Controllable Video Generation

## Abstract

To address the bottleneck of accurate user intent interpretation within current video generation community, we present `Any2Caption`, a novel framework for controllable video generation from any condition. The key idea is decoupling various condition interpretation steps from the video synthesis step. By leveraging modern multimodal large language models (MLLMs), Any2Caption interprets diverse inputs—text, images, videos, and specialized cues such as region, motion, and camera poses—into dense, structured captions that offer backbone video generators with better guidance. We also introduce `Any2CapIns`, a large-scale dataset with 337K instances and 407K conditions for any-condition-to-caption instruction tuning. Comprehensive evaluations demonstrate significant improvements in our system's controllability and video quality compared to existing video generation models. Codes and resources will be released upon acceptance.

## 1 Introduction

Video serves as a fundamental medium for capturing real-world dynamics, making diverse and controllable video generation a key capability for modern artificial intelligence (AI) systems. Recently, video generation has gained significant attention, driven by advancements in Diffusion Transformers (DiT) Zheng et al. (2024b); Jiang et al. (2024); Singer et al. (2023); Ma et al. (2024a); kua (2024), which have demonstrated the ability to generate realistic, long-duration videos from text prompts. These advancements have even led to industrial applications, such as filmmaking. However, we observe that a major bottleneck in the current video generation community lies in **accurately interpreting user intention**, so as to produce high-quality, controllable videos.

In text-to-video (T2V) generation, studies Yang et al. (2024c); Ju et al. (2024); Hong et al. (2023) have suggested that detailed prompts, specifying objects, actions, attributes, poses, camera movements, and style, significantly enhance both controllability and video quality. Thus, a series of works have explored video recaption techniques (e.g., ShareGPT4Video Chen et al. (2024a), MiraData Ju et al. (2024), and InstanceCap Fan et al. (2024)) to build dense structural captions for optimizing generation models. While dense captions are used during training, in real-world inference scenarios, users most likely provide concise or straightforward input prompts Fan et al. (2024). Such a gap inevitably weakens instruction following and leads to suboptimal generation due to an incomplete understanding of user intent. To combat this, there are two possible solutions, manual prompt refinement, or automatic prompt enrichment Fan et al. (2024); Yang et al. (2024c) using large language models (LLMs). Yet, these approaches either require substantial human effort or risk introducing noise from incorrect prompt interpretations. As a result, this limitation in precisely interpreting user intent hinders the adoption of controllable video generation for demanding applications such as anime creation and filmmaking.

On the other hand, to achieve more fine-grained controllable video generation, one effective strategy is to provide additional visual conditions besides text prompts—such as reference images Wu et al. (2023); Guo et al. (2024a), identity Yuan et al. (2024); He et al. (2024b); Ma et al. (2024c), style Ye et al. (2024); Liu et al. (2024a), human pose Ma et al. (2024b); Karras et al. (2023), or camera He et al. (2024a); Zheng et al. (2024a)—or even combinations of multiple conditions together Zhao et al. (2023); Lin et al. (2024b); Wang et al. (2023). This multimodal conditioning approach aligns well with real-world scenarios, as users quite prefer interactive ways to articulate their creative intent. Several studies have examined video generation under various conditions, such

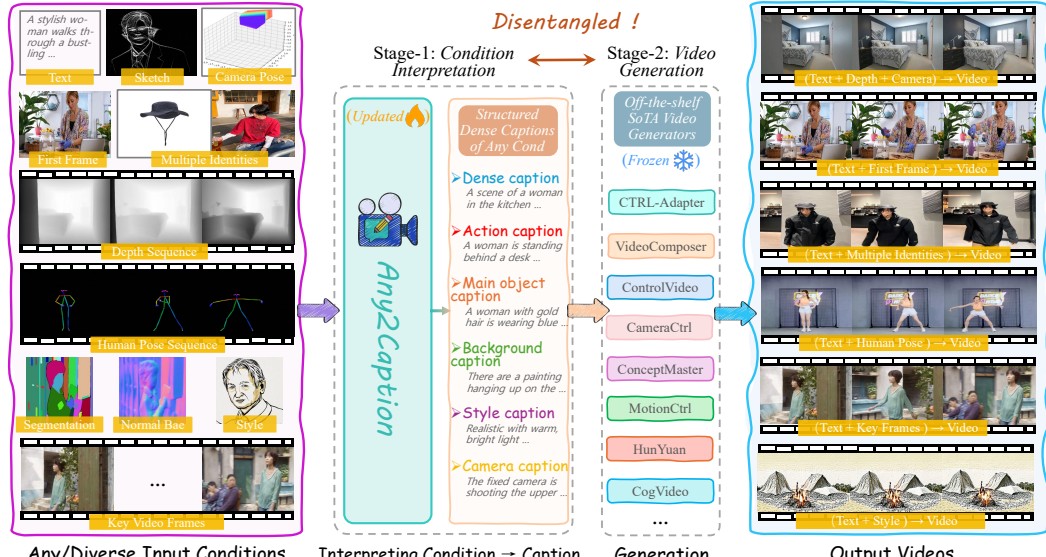

Figure 1: We propose `Any2Caption`, an efficient and versatile framework for interpreting diverse conditions to structured captions for highly controllable video generation.

as VideoComposer Wang et al. (2023), Ctrl-Adapter Lin et al. (2024b), and ControlVideo Zhao et al. (2023). Unfortunately, these methods tend to rely on the internal encoders of Diffusion/DiT to parse rich heterogeneous input conditions with intricate requirements (e.g., multiple object IDs and complex camera movements). Before generation, the model must accurately interpret the semantics of varied visual conditions in tandem with textual prompts. Yet even state-of-the-art (SoTA) DiT backbones have limited capacity for reasoning across different input modalities, resulting in suboptimal generation quality.

This work is dedicated to addressing these bottlenecks of any-conditioned video generation. Our core idea is to **decouple the first job of interpreting various conditions from the second job of video generation**, motivated by two important observations: **a)** SoTA video generation models (e.g., DiT) already excel at producing high-quality videos when presented with sufficiently rich text captions; **b)** Current MLLMs have demonstrated robust vision-language comprehension. Based on these, we propose **Any2Caption**, an MLLM-based universal condition interpreter designed not only to handle text, image, and video inputs but also equipped with specialized modules for motion and camera pose inputs. As illustrated in Fig. 1, `Any2Caption` takes as inputs any conditions (or combinations) and produces a densely structured caption, which is then passed on to any backend video generator for controllable, high-quality video production. As `Any2Caption` disentangles the role of complex interpretation of multimodal inputs from the generator, it advances in seamlessly integrating into a wide range of well-trained video generators without the extra cost of fine-tuning.

To facilitate the any-to-caption instruction tuning for `Any2Caption`, we construct **Any2CapIns**, a large-scale dataset that converts a concise user prompt and diverse non-text conditions into detailed, structured captions. Concretely, the dataset encompasses four main categories of conditions: depth maps, multiple identities, human poses, and camera motions. Through extensive manual labeling combined with automated annotation by GPT-4V Gpt (2023), followed by rigorous human verification, we curate a total of **337K** high-quality instances, with **407K** condition annotations, with the short prompts and structured captions averaging 55 and 231 words, respectively. Additionally, we develop a comprehensive evaluation strategy to thoroughly assess the model's ability to interpret user intent under various conditions.

Experimentally, we first validate `Any2Caption` on our `Any2CapIns`, where results demonstrate that it achieves an impressive captioning quality that can faithfully reflect the original input conditions. We then experiment with integrating `Any2Caption` with multiple SoTA video generators, finding that (a) the long-form semantically rich prompts produced by `Any2Caption` are pivotal for generating high-quality videos under arbitrary conditions, and (b) `Any2Caption` consistently enhances performance across different backbone models, yielding noticeably improved outputs. Furthermore, `Any2Caption` demonstrates a pronounced advantage when handling mul-

tiple combined conditions, effectively interpreting and synthesizing intricate user constraints into captions that closely align with user expectations. Our contributions are threefold:

- We for the first time pioneer a novel *any-condition-to-caption* paradigm for video generation, which bridges the gap between user-provided multimodal conditions and structured video generation instructions, leading to highly controllable video generation.
- We propose `Any2Caption` to effectively integrate and comprehend diverse multimodal conditions, producing semantically enriched and structured captions, which consistently improve both condition flexibility and video quality. `Any2Caption` can also be widely integrated as a plug-in module for any existing video generator.
- We introduce `Any2CapIns`, a large-scale, high-quality dataset for any-condition-to-caption task, and establish a suite of evaluation metrics to rigorously assess the quality and fidelity of condition-based caption generation.

## 2 RELATED WORK

Controllable video generation Sun et al. (2024); Chen et al. (2024d); Fang et al. (2024); He et al. (2024a) has long been a central topic in AI. Recent advanced DiT methods, such as OpenAI's Sora sor (2024) and HunyuanVideo Kong et al. (2024), yield photorealistic videos over extended durations. Early work focused on text-controlled video generation Singer et al. (2023); Hong et al. (2023), the prevalent approach. Yet, text prompts alone may insufficiently capture user intent, spurring exploration of additional inputs including static images Wu et al. (2023); Guo et al. (2024a), sketches Zhao et al. (2023); Wang et al. (2023), human poses Zhong et al. (2024); Ma et al. (2024b); Karras et al. (2023), camera views Zheng et al. (2024a); He et al. (2024a), and even extra videos Kara et al. (2024); Deng et al. (2024); Zhang et al. (2023). Thus, unifying these diverse conditions into an "any-condition" framework is highly valuable.

Recent works such as VideoComposer Wang et al. (2023), Ctrl-Adapter Lin et al. (2024b), and ControlVideo Zhao et al. (2023) have explored any-condition video generation. However, they face challenges in controlling multiple modalities due to the limited interpretability of text encoders in Diffusion or DiT. Motivated by existing MLLMs' multimodal reasoning Liu et al. (2024b); Lin et al. (2024a); Wang et al. (2024a), we propose leveraging an MLLM to consolidate all possible conditions into structured dense captions for better controllable generation. SoTA DiT models already exhibit the capacity to interpret dense textual descriptions as long as the input captions are sufficiently detailed in depicting both the scene and the intended generation goals. Thus, our MLLM-based encoder alleviates the comprehension bottleneck, enabling the generation of higher-quality videos. To our knowledge, this is the first attempt in the field of any-condition video generation. Moreover, as the captioning stage is decoupled from backbone DiT, `Any2Caption` can integrate with existing video generation solutions without additional retraining.

Our approach also relates to video recaptioning, as our system generates dense captions under specific conditions. In text-to-video settings, prior work Fan et al. (2024); Yang et al. (2024c); Nan et al. (2024); Islam et al. (2024) shows that recaptioning yields detailed annotations that improve DiT training. For instance, ShareGPT4Video Chen et al. (2024a) uses GPT-4V Gpt (2023) to reinterpret video content, while MiraData Ju et al. (2024) and InstanceCap Fan et al. (2024) focus on structured and instance-consistent recaptioning. Unlike these methods, we avoid retraining powerful DiT models with dense captions by training an MLLM as an any-condition encoder on pairs of short, dense captions that are easier to obtain. Moreover, recaptioning entire videos can introduce noise or hallucinations that undermine DiT training, whereas our framework sidesteps this risk. Finally, while previous studies rely on dense-caption-trained DiT models, the real-world user concise prompts might create a mismatch that degrades generation quality.

## 3 ANY2CAPINS DATASET CONSTRUCTION

While relevant studies recapturing target videos for dense captions for enhanced T2V generation Chen et al. (2024a); Fan et al. (2024); Ju et al. (2024), these datasets suffer from two key limitations: **1)** the absence of non-text conditions, and **2)** short prompts that do not account for interactions among non-text conditions, potentially leading to discrepancies in real-world applications. To address these limitations, we introduce a new dataset, `Any2CapIns`, specifically designed to

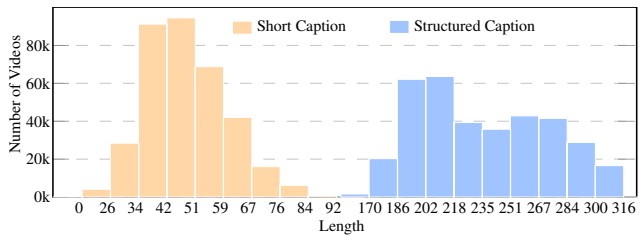

Figure 2: The pipeline for constructing the `Any2CapIns` dataset involves three key steps: 1) data collection, 2) structured video caption generation, and 3) user-centric short prompt generation.

incorporate diverse multimodal conditions for generating structured video captions. The dataset is constructed through a three-step process (cf. Fig. 2).

**Step-1: Data Collection.** We begin by systematically categorizing conditions into four primary types: 1) **Spatial-wise conditions**, focus on the structural and spatial properties of the video, e.g., depth maps, sketches, and video frames. 2) **Action-wise conditions**, emphasize motion and human dynamics in the target video, e.g., human pose motion. 3) **Composition-wise conditions**, focus on scene composition, particularly in terms of object interactions and multiple identities

| Category | #Num. | #Condition | #Avg. Len. | #Total Len. |
|---|---|---|---|---|
| Depth | 182,945 | 182,945 | 9.87s | 501.44h |
| Human Pose | 44,644 | 44,644 | 8.38s | 108.22h |
| Multi-Identities | 68,255 | 138,089 | 13.01s | 246.69h |
| Camera Movement | 41,112 | 41,112 | 6.89s | 78.86h |

Table 1: Statistics of the collected dataset across four types of conditions. **#Num. / #Condition** means the number of instances / unique conditions. **#Avg. / #Total Len.** indicate the average and total video durations, respectively.

in the target video. 4) **Camera-wise conditions**, control video generation from a cinematographic perspective, e.g., movement trajectories. Since it is infeasible to encompass all possible conditions in dataset collection, we curate representative datasets Zhou et al. (2018); Chen et al. (2024c); Wang et al. (2024b); Ma et al. (2024b) under each type, specifically including *depth maps*, *human pose*, *multiple identities*, and *camera motion*. During the data collection process, we leverage SoTA tools to construct conditions. For instance, Depth Anything Yang et al. (2024b) is used to generate depth maps, DW-Pose Yang et al. (2023) provides human pose annotations, and Sam2 Ravi et al. (2024) is utilized for segmentation construction. In total, we collect **337K** video instances and **407K** conditions, with detailed statistics of the dataset presented in Tab. 1.

**Step-2: Structured Video Caption Generation.** The granularity of a caption, i.e., the specific elements it encompasses, plays a critical role in guiding the model to produce videos that closely align with desired descriptions while preserving coherence and realism. Drawing inspiration from Ju et al. (2024); Liu et al. (2024d), we design a structured caption format consisting of (1) `Dense caption`, (2) `Main object caption`, (3) `Background caption`, (4) `Action caption`, (5) `Style caption`, and (6) `Camera caption`. Following Wang et al. (2024b), we leverage GPT-4V and a fine-tuned LLaVA to generate the aspects of structured caption separately and merge them into a final structured caption.

**Step-3: User-centric Short Prompt Generation.** In this step, we construct short prompts from a user-centric perspective, considering how users naturally express their intentions. Firstly, our analysis highlights three key characteristics of users prompts: 1) **Conciseness and Simplicity**: Users favor brief and straightforward wording; 2) **Condition-Dependent Omission**: Users often omit textual descriptions of certain attributes (e.g., camera movement) when such conditions are already specified; and 3) **Implicit instruction of Target Video**: Users convey their intent indirectly (e.g., specifying multiple identities without detailing

Figure 3: **Distribution of the short/structured caption length** (in words) in `Any2CapIns`.

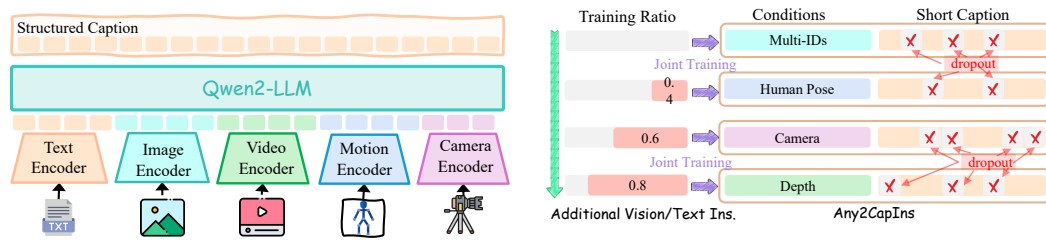

(a) Overall architecture of Any2Caption      (b) Progressive Mixed Training

Figure 4: The Architecture of `Any2Caption` (a) uses Qwen2-LLM as the backbone, paired with text, image, video, motion, and camera encoders to generate structured captions. After alignment learning, a progressive mixed training strategy (b) is employed, progressively adding vision/text instruction datasets for joint training. For input short captions, a sentence-level random dropout mechanism is used to enhance robustness.

interactions). Guided by these observations, we tailor GPT-4V to precisely infer potential user prompts under condition-specific constraints. We also explicitly control the prompt length to maintain conciseness and refine the dataset through manual verification and filtering. Fig. 3 presents the length distribution of the resulting short and structured prompts.

# 4 ANY2CAPTION MODEL

In this section, we present `Any2Caption`, a novel MLLM designed to comprehensively interpret arbitrary multimodal conditions into controllable video captions, as illustrated in Fig. 4(a). Formally, given a short text prompt $T$ along with non-text conditions $C = [c_1, \cdots, c_n]$, where the non-text conditions can be either none, a single condition, or multiple conditions. The objective of this task is to generate a detailed and structured caption that serves as a control signal for video generation.

**Architecture.** Similar to existing MLLMs Liu et al. (2024b); Wu et al. (2024); Lin et al. (2024a), `Any2Caption` incorporates an image encoder $F_{\mathcal{I}}$, a video encoder $F_{\mathcal{V}}$, a motion encoder $F_{\mathcal{M}}$ and a camera encoder $F_{\mathcal{C}}\}$ to process non-text conditions. These encoders are then integrated into an LLM backbone $F_{\mathcal{LLM}}$ (i.e., Qwen2-LLM) to facilitate structured video captioning. Specifically, we leverage a ViT-based visual encoder from Qwen2-VL as $F_{\mathcal{I}}$ and $F_{\mathcal{V}}$ for the unified modeling of images and videos, achieving effective interpretation of input conditions represented in image or video formats, such as depth maps and multiple identities. To enable human pose understanding, we represent the extracted human pose trajectories as $\boldsymbol{H}=\{(x_n^k, y_n^k)|k=1, \cdots, K, n=1, \cdots, N\}$, where $N$ denotes the number of video frames and $K$ is the number of keypoints. These trajectories are then visualized within video frames to enable further processing by the motion encoder, which shares the same architectural structure and initialization as the vision encoder. For camera motion understanding, inspired by He et al. (2024a), we introduce a camera encoder that processes a plücker embedding sequence $P \in \mathbb{R}^{N \times 6 \times H \times W}$, where $H$, $W$ are the height and width of the video. This embedding accurately captures camera pose information, enabling precise modeling of camera trajectories. Finally, in line with Qwen2-VL, we employ special tokens to distinguish non-text conditions from texts. Besides the existing tokens, we introduce `<|motion_start|>`, `<|motion_end|>`, `<|camera_start|>`, `<|camera_end|>`, to demarcate the start and end of human&camera pose features.

**Training Recipes.** To accurately interpret user generation intent under arbitrary conditions and yield structured target video captions, large-scale pretraining and instruction tuning are required. To this end, we adopt a two-stage training procedure: **Stage-I: Alignment Learning.** In this stage, as image and video encoders have been well-trained in Qwen2-VL, we only focus on aligning human pose features from the motion encoder and camera movement features with the word embeddings of the LLM backbone. To achieve this, we freeze the LLM and vision encoder, while keeping the motion encoder trainable and optimize it on a human pose description task. Similarly, for camera movement alignment, we unfreeze the camera encoder and train it on a camera movement description task, ensuring that camera-related conditions are embedded into the model's latent space. This alignment phase establishes a strong foundation for effective representation learning for these conditions. **Stage II: Condition-Interpreting Learning** Building upon the aligned encoders and

pretrained Qwen2-VL weights, we fine-tune the model on Any2CapIns for multimodal condition interpretation. However, direct fine-tuning leads to catastrophic forgetting due to the fixed output structure. To address this, we propose a progressive mixed training strategy. Specifically, the model is first trained on a single condition to establish a strong condition-specific understanding. As new conditions are introduced, we gradually incorporate vision-language instructions such as LLaVA-instruction Liu et al. (2024b) and Alpaca-52K Taori et al. (2023). This stepwise strategy ensures robust multimodal condition interpretation while preventing knowledge degradation.

## 5 EVALUATION SUITE

In this section, we introduce the evaluation suite for comprehensively assessing the capability of `Any2Caption` in interpreting user intent and generating structured captions.

**Lexical Matching Score.** We employ standard evaluation metrics commonly used in image/video captioning tasks, including `BLEU` Papineni et al. (2002), `ROUGE` Lin (2004), and `METEOR` Banerjee & Lavie (2005). We also introduce a `Structural Integrity` score to verify whether the generated captions adhere to the required six-component format, thereby ensuring completeness.

**Semantic Matching Score.** To evaluate the semantic alignment of generated captions, we employ BERTSCORE Zhang et al. (2020), which computes similarity by summing the cosine similarities between token embeddings, effectively capturing both lexical and compositional meaning preservation. Additionally, we utilize CLIP Score Hessel et al. (2021) to assess the semantic consistency between the input visual condition and the generated videos.

**Intent Reasoning Score.** Inspired by Chai et al. (2025), we introduce the Intent Reasoning Score (IRSCORE) to assess the fidelity of generated captions that capture user intents. The IRSCORE evaluation framework involves four steps: **(1) User Intention Extraction:** Categorize user intent into six aspects: subject, background, movement, camera, interaction, and style. **(2) Gold QA Pair Construction:** Create aspect-specific QA pairs with defined requirements (e.g., object count, appearance). **(3) Answer Prediction:** Prompt GPT-4V to answer the questions solely based on the predicted caption. **(4) Answer Evaluation:** GPT-4V scores each answer for correctness and quality, averaged across all QA pairs. More details are in Appendix §G.

**Video Generation Quality Score.** We also employ several metrics to evaluate the quality of videos generated from structured captions. Following Huang et al. (2024); Ju et al. (2024), we evaluate video generation quality across four key dimensions: motion `smoothness`, `dynamic degree`, `aesthetic` quality, and image `integrity`. To further verify adherence to specific non-text conditions, we use specialized metrics: `RotErr`, `TransErr`, and `CamMC` He et al. (2024a) for camera motion accuracy; `Mean Absolute Error (MAE)` for depth consistency Guo et al. (2024b); `DINO-I` Ruiz et al. (2023), `CLIP-I` Ruiz et al. (2023) Score to evaluate identity preservation under multiple identities, and Pose Accuracy (`Pose Acc.`) Ma et al. (2024b) to access the alignment in the generated videos.

## 6 EXPERIMENTS

### 6.1 SETUPS

**Dataset.** We manually construct 200 test cases for each type of condition (i.e., *depth*, *human pose*, *multiple identities*, *camera*, and *compositional conditions*) to evaluate the model's performance. Additionally, we assess the model on publicly available benchmarks (e.g., Ye et al. (2024); Lin et al. (2024b)). For further details, please refer to the Appendix §H.

**Implementation Details.** We leverage Qwen2VL-7B Wang et al. (2024a) as the backbone of our model, which supports both image and video understanding. The human pose in the input conditions is encoded and processed in the video format. The camera encoder adopts the vision encoder architecture, with the following settings: in channels set to 96, patch size of 16, depth of 8, and 8 attention heads. During training, to simulate the brevity and randomness of user inputs, we randomly drop sentences from the short caption with a dropout rate of 0.6; a similar dropout strategy is applied to non-textual conditions. We conducted the training on 8×A800 GPUs. For further details on the training parameters for each stage, please refer to the Appendix §H.

| Category | Structural Integrity | Lexical Matching | | | Semantic Matching | Intent Reasoning | |
| --- | --- | --- | --- | --- | --- | --- | --- |
| | | B-2 | R-L | METER | BERTSCORE | Accuracy | Quality |
| Entire Structured Caption | 91.25 | 54.99 | 48.63 | 52.47 | 91.95 | 68.15 | 3.43 |
| Dense Caption | - | 44.24 | 42.89 | 49.51 | 92.42 | 78.47 | 3.47 |
| Main Object Caption | - | 38.54 | 47.46 | 52.48 | 92.02 | 56.28 | 2.74 |
| Background Caption | - | 44.65 | 46.73 | 48.87 | 92.90 | 69.37 | 2.69 |
| Action Caption | - | 31.91 | 39.83 | 45.25 | 91.44 | 57.98 | 2.13 |
| Style Caption | - | 41.71 | 47.70 | 55.9 | 93.48 | 63.75 | 3.05 |
| Camera Caption | - | 60.21 | 96.10 | 94.32 | 99.31 | 66.31 | 3.75 |

Table 4: Quantitative results of structured caption generation quality under four aspects: *structural Integrity*, *lexical matching*, *semantic matching*, and *intent reasoning*. We demonstrate the overall caption generation capability and the performance of individual components within the structure. "B-2" and "R-L" denotes BLEU-2 and ROUGE-L, respectively.

## 6.2 EXPERIMENTAL RESULTS AND ANALYSES

In this section, we present the experimental results and provide in-depth analyses to answer the following six key research questions, revealing how the system advances.

**RQ-1: Is the structured caption necessary?** We compare our structured caption approach with a simpler method, where we first caption the input condition (e.g., multiple identity images) and then concatenate that caption with the original short prompt, as shown in Tab. 2. Our results indicate that merely appending the condition's caption to the short prompt can reduce video smoothness

| Prompt Enrichment | Text | Video Generation | | |
| --- | --- | --- | --- | --- |
| | DINO-I↑ | Smoo.↑ | Aest.↑ | Inte.↑ |
| Multi-IDs+Prompt. | 43.72 | 93.46 | 5.32 | 55.39 |
| Multi-IDs+Prompt+IDs Cap. | 43.96 | 93.41 | 5.41 | 54.91 |
| Multi-IDs+Structured Cap. | **45.77** | **94.38** | **5.46** | **57.47** |
| Only Structured Cap. | 38.45 | 94.43 | 5.19 | 56.09 |

Table 2: Quantitative results comparing multi-identity video generation using different prompt enrichment methods.

and image quality. One likely reason is that the identity images may contain extraneous details beyond the target subject, potentially conflicting with the original prompt and causing inconsistencies. Consequently, controllability in the final output is compromised. In contrast, our structured caption method accurately identifies the target subject and augments the prompt with relevant information, yielding more controllable video generation (cf. §I.3).

**RQ-2: How effective is the training strategy?** Next, we investigate the contribution of the training mechanism, and the results are shown in Tab. 3. During training, we employ a two-stage training approach, consisting of alignment learning followed by instruction-tuning. When alignment learning is omitted, and the model proceeds directly to instruction tuning, both captioning and video gener-

| Training Strategy | Caption | | Vieo Generation | | |
| --- | --- | --- | --- | --- | --- |
| | B-2↑ | Accuracy↑ | Smoo.↑ | Dyna.↑ | Aest.↑ |
| Any2Caption | 47.69 | 67.35 | **94.60** | **17.67** | **5.53** |
| w/o Two-Stage | 33.70 | 51.79 | 93.31 | 16.36 | 5.10 |
| w/o Dropout | **49.24** | **69.51** | 94.16 | 14.54 | 5.51 |

Table 3: Ablation study on training strategy. "w/o Two-stage" means no alignment learning, and "w/o Dropout" denotes short captions are not randomly dropped.

ation performance degrade significantly. A possible explanation is that bypassing alignment learning disrupts the encoder's adaptation process which has been aligned to the LLM backbone, leading to suboptimal results in subsequent stages. Additionally, we compare the performance of the model without the dropout mechanism. Although removing dropout yields a marked improvement in captioning quality, the benefit to video generation is marginal. This suggests that without dropout, the model may rely on shortcuts from the input captions rather than fully understanding the underlying intent, thereby increasing the risk of overfitting.

**RQ-3: How well is the structured caption generation quality?** We first evaluate whether our proposed model could accurately interpret user intent and generate high-quality structured captions. From a caption-generation perspective, we compare the predicted captions with gold-standard captions across various metrics (see Tab. 4). We observe that our model successfully produces the desired structured content, achieving 91.25% in structural integrity. Moreover, it effectively captures

**Example-1**: A man gestures while the woman listens. They sit in a sunny park. The camera captures close-up shots of their heads and shoulders.

Short Cap. + IDs → *Any2Caption* → Structured Cap. → *CogVideoX-2B* → Video

**Example-2**: A serene winter backyard with snow-covered ground and bare trees, revealing a blue shed with a white garage door and a doghouse

Short Cap. + Camera + Depth → *Any2Caption* → Structured Cap. → *CogVideoX-2B* → Video

**Example-3**: A young man carrying a messenger bag runs down a narrow, cobblestone street filled with sandbags and crates, suggesting a wartime.

Short Cap. + ID → *Any2Caption* → Structured Cap. → *HunyuanVideo* → Video

**Example-4**: A woman walks in a minimalist, modern room. She is holding two mugs and looks slightly displeased. The room has natural light.

Short Cap. + IDs + Depth → *Any2Caption* → Structured Cap. → *HunyuanVideo* → Video

Figure 5: Illustrations of generated videos where only the structured captions yielded by *Any2Caption* are fed into the *CogVideoX-2B* (Left), and *HunyuanVideo* (Right). We can observe that some key features of the input identity images, such as the background and main object, can be accurately visualized in the generated videos.

| Model | Text | Camera | | | Identities | | Depth | Human Pose | Overall Quality | | |
|---|---|---|---|---|---|---|---|---|---|---|---|
| | CLIP-T↑ | RotErr↓ | TransErr↓ | CamMC↓ | DINO-I↑ | CLIP-I↑ | MAE↓ | Pose Acc.↑ | Smoo.↑ | Dyna.↑ | Inte.↑ |
| ● **Camera to Video** | | | | | | | | | | | |
| MotionCtrl | 19.67 | 1.54 | 4.49 | 4.80 | - | - | - | - | 96.13 | 9.75 | 73.69 |
| + Structured Cap. | **20.16** | **1.45** | **4.37** | **4.78** | - | - | - | - | **96.16** | **11.43** | **74.63** |
| CameraCtrl | 18.89 | 1.37 | 3.51 | 4.78 | - | - | - | - | 94.11 | 12.59 | 71.84 |
| + Structured Cap. | **21.70** | **0.94** | **2.97** | **4.37** | - | - | - | - | **95.16** | **13.72** | **72.47** |
| ● **Depth to Video** | | | | | | | | | | | |
| Ctrl-Adapter | 20.37 | - | - | - | - | - | 25.63 | - | 94.53 | **20.73** | 46.98 |
| + Structured Cap. | **23.30** | - | - | - | - | - | **21.87** | - | **95.54** | 15.14 | **54.20** |
| ControlVideo | 22.17 | - | - | - | - | - | 30.11 | - | 92.88 | 5.94 | 63.85 |
| + Structured Cap. | **24.18** | - | - | - | - | - | **23.92** | - | **94.47** | **18.27** | **66.28** |
| ● **Identities to Video** | | | | | | | | | | | |
| ConceptMaster | 16.04 | - | - | - | 36.37 | 65.31 | - | - | 94.71 | 8.18 | 43.68 |
| + Structured Cap. | **17.15** | - | - | - | **39.42** | **66.74** | - | - | **95.05** | **10.14** | **49.73** |
| ● **Human Pose to Video** | | | | | | | | | | | |
| FollowYourPose | 21.11 | - | - | - | - | - | - | 30.47 | 91.71 | 14.29 | **58.84** |
| + Structured Cap. | **21.39** | - | - | - | - | - | - | **31.59** | **92.87** | **16.47** | 56.30 |

Table 5: Performance on *camera*, *depth*, *identities*, and *human pose* conditions for using short vs. structured captions, evaluated using various video quality metrics. Better results are in **bold**.

the key elements of the gold captions, attaining a ROUGE-L score of 48.63 and a BERTSCORE of 91.95. Notably, the model demonstrates the strongest performance in interpreting camera-related details compared to other aspects. Finally, regarding user intent analysis, we found that the model reliably incorporated user preferences into its structured outputs.

To further showcase the model's capacity to understand and leverage input conditions, we directly feed the structured captions—derived from our model's interpretation—into downstream text-to-video generation systems (e.g., CogvideoX Yang et al. (2024c) and Hunyuan Kong et al. (2024)), as illustrated in Fig. 5. Even without explicit visual conditions (e.g., identities), the resulting videos align well with the input prompts, such as hair color and clothing style in Example 1, indicating that our captions successfully capture intricate visual details. In particular, the model is able to accurately grasp dense conditions, such as depth sequences, or compositional requirements in Example 4, ultimately enabling controllable video generation. Although some subtle attributes may be under-specified in text, occasionally leading to mismatches, overall controllability remains robust.

**RQ-4: Is the video generation quality enhanced with a structured caption?** Here, we investigate whether integrating structured captions consistently improves controllable video generation in multiple methods. We investigate the impact of incorporating camera, depth, identities, and human pose conditions into various controllable video generation methods. As shown in Table 5, all tested models exhibit consistent gains in overall video quality, including smoothness and frame fidelity, after incorporating structured captions, without requiring any changes to the model architectures or additional training. Moreover, these models show enhanced adherence to the specified conditions, suggesting that our generated captions precisely capture user requirements and lead to more accurate, visually coherent video outputs. Additional examples can be found in Appendix §I.3.

| Conditions | Text | Camera | | | Identities | | Depth | Overall Quality | | | |
|---|---|---|---|---|---|---|---|---|---|---|---|
| | CLIP-T↑ | RotErr↓ | TransErr↓ | CamMC↓ | DINO-I↑ | CLIP-I↑ | MAE↓ | Smoo.↑ | Dyna.↑ | Aest.↑ | Inte.↑ |
| C+I | 14.81 | 1.37 | **4.04** | 4.24 | 25.63 | 64.14 | - | **94.43** | 28.87 | 4.99 | 59.81 |
| + Structured Cap. | **19.03** | **1.30** | 4.36 | **4.03** | **26.75** | **68.45** | - | 94.38 | **34.99** | **5.25** | **63.02** |
| C+D | 20.80 | 1.57 | **3.88** | 4.77 | - | - | 32.15 | 95.36 | **30.12** | 4.82 | 63.90 |
| + Structured Cap. | **21.19** | **1.49** | 4.41 | 4.84 | - | - | **25.37** | **95.40** | 30.10 | **4.96** | **65.05** |
| D+I | 20.01 | - | - | - | 35.24 | 57.82 | **23.00** | **93.15** | 32.21 | 4.96 | **61.21** |
| + Structured Cap. | **20.76** | - | - | - | **36.25** | **63.48** | 24.78 | 92.50 | **36.43** | **5.18** | 60.81 |
| C+I+D | 18.49 | 2.05 | **7.74** | 8.47 | 35.86 | 64.25 | 18.37 | 92.02 | 30.09 | 3.91 | 60.62 |
| + Structured Cap. | **19.52** | **1.57** | **7.74** | **8.20** | **38.74** | **64.37** | **17.41** | **93.03** | **32.81** | **4.99** | **61.22** |

Table 6: Quantitative comparison across compositional conditions on FullDiT Ju et al. (2025). C, D, and I denote `camera`, `depth` and `multiple identities` conditions, respectively

**RQ-5: How well does the model perform on compositional conditions?** We examine the impact of structured captions under compositional conditions. As shown in Tab. 6, we compare the combined camera, identities, and depth on our customized model and observe that structured captions consistently enhance its performance. Moreover, from Examples 2 and 4 in Fig. 5, our model demonstrates a thorough understanding of the interactions among various conditions, for instance, capturing a woman's hair color and the position of a mug, accurately guiding the production of videos that align with the specified requirements. This finding further highlights that our approach can automatically equip existing T2V models with the ability to handle compositional conditions without requiring additional training.

**RQ-6: How well is the generalization capability of Any2Caption?** Finally, we investigate the model's generalization ability by evaluating its performance on "unseen" conditions, including *style*, *segmentation*, *sketch*, and *masked images*. As demonstrated in Fig. 6, the structured captions generated by our model consistently enhance existing T2V frameworks, offering benefits such as increased motion smoothness, aesthetics quality, and more accurate generation control. We attribute these advantages to two primary factors: the strong reasoning capabilities of our MLLM backbone and our training strategy, i.e., progressive mixed training, which leverages existing vision and text instructions for fine-tuning while mitigating knowledge forgetting, thereby ensuring robust generalization.

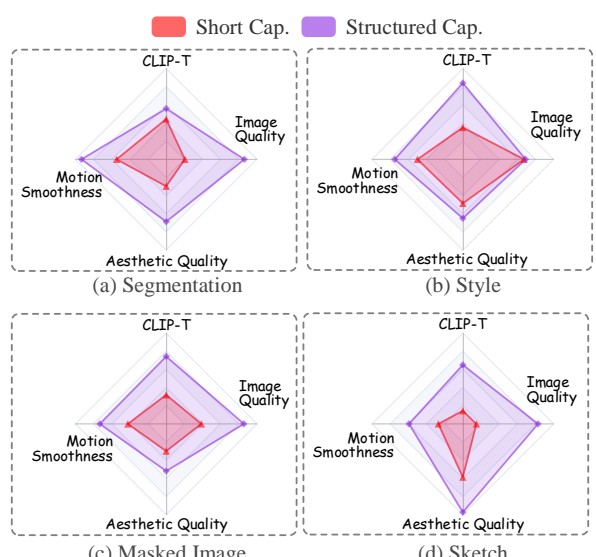

Figure 6: Quantitative results on unseen conditions (i.e., *segmentation* Wang et al. (2023), *style* Ye et al. (2024), *masked image* Wang et al. (2023), and *sketch* Wang et al. (2023)) when using short and structured captions, respectively.

## 7 CONCLUSION

In this work, we focus on addressing the challenge of accurately interpreting user generation intentions under various conditions for controllable video generation. We introduce `Any2Caption`, a framework that decouples the interpretation of multimodal conditions from video synthesis. Built based on an MLLM, `Any2Caption` converts diverse inputs into dense captions that drive high-quality video generation. We further present `Any2CapIns`, a large-scale dataset for effective instruction tuning. Experiments show that our method improves controllability and video quality across various backbones.

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

APPENDIX OVERVIEW

The appendix presents more details and additional results not included in the main paper due to page limitations. The list of items included is:

- Clarification on the Usage of Large Language Models in Section §A.
- Limitation in Section §B.
- Ethic Statement results in Section §C.
- Reproducibility Statement in Section §D.
- Extended Related Work in Section §E.
- Extended Dataset Construction Details in Section §F.
- More Statistics Information of IRSCORE in Section §G.
- Detailed Setups in Section §H.
- Extended Experiment Results and Analyses in Section §I.

## A  THE USAGE OF LARGE LANGUAGE MODELS (LLMs)

In this paper, we employed GPT-4o for data construction and, separately, leveraged large language models (LLMs) as auxiliary tools to improve the clarity and readability of the manuscript. Specifically, LLMs were utilized to refine sentence structure, correct grammatical errors, and enhance the overall presentation of the draft. All technical content, including research ideas, algorithm design, experimental setup, analysis, and conclusions, was entirely conceived, implemented, and validated by the authors without reliance on LLMs.

## B  LIMITATION

Despite the advancement of our proposed framework, several limitations may remain:

Firstly, the diversity of annotated data is constrained by the capabilities of the current annotation tools, which may limit the variety of generated content. Moreover, the scarcity of real-world data introduces potential domain gaps, reducing the model's generalizability in practical scenarios.

Secondly, due to inherent model limitations, hallucinations may occur, resulting in inaccurate structured captions and consequently degrading the quality of generated videos. A possible direction to mitigate this issue is to develop an end-to-end approach that jointly interprets complex conditions and handles video generation.

Lastly, the additional condition-understanding modules inevitably increase inference time. However, our empirical results suggest that the performance gains from these modules are substantial, and future work may explore more efficient architectures or optimization techniques to balance speed and accuracy.

## C  ETHIC STATEMENT

This work relies on publicly available datasets and manually constructed datasets, ensuring that all data collection and usage adhere to established privacy standards. We recognize that automatic annotation processes may introduce biases, and we have taken measures to evaluate and mitigate these biases. Nonetheless, we remain committed to ongoing improvements in this area.

By enhancing video generation capabilities, `Any2Caption` could inadvertently facilitate negative societal impacts, such as the production of deepfakes and misinformation, breaches of privacy, or the creation of harmful content. We, therefore, emphasize the importance of strict ethical guidelines, robust privacy safeguards, and careful dataset curation to minimize these risks and promote responsible research practices.

## D    REPRODUCIBILITY STATEMENT

To ensure the reproducibility of our work, we have made a concerted effort to provide all necessary details and materials. We provide comprehensive details of the proposed Any2Caption framework, including its definition and input–output formulation (Section §4). The model backbone and training methodology are described in detail in Section §4 and Appendix §H.3. We further report all hyperparameter settings and training configurations in Section §6.1 and Appendix §H, using fixed random seeds to ensure the replicability of the experiments. All datasets used in this study are publicly available open-source resources, and the data construction process, along with the amount of data used at each training stage, is thoroughly documented in Section §3 and Appendix §F. Finally, we will release the full codebase and data processing scripts to the community upon acceptance.

## E    EXTENDED RELATED WORK

### E.1    TEXT-TO-VIDEO GENERATION

The development of video generation models has progressed from early GAN- and VAE-based approaches Brooks et al. (2022); Wang et al. (2020); Haim et al. (2022); Chu et al. (2020); Gur et al. (2020) to the increasingly popular diffusion-based methods Blattmann et al. (2023b;a); Zhang et al. (2024c); Qing et al. (2024). Among these, diffusion-in-transformer (DiT) architectures, such as OpenAI's Sora sor (2024) and HunyuanVideo Kong et al. (2024), have demonstrated remarkable performance, producing photorealistic videos over extended durations. Controllable video generation Sun et al. (2024); Chen et al. (2024d); Fang et al. (2024); He et al. (2024a) has become an essential aspect of this field. Initially, research efforts centered predominantly on text-to-video generation Singer et al. (2023); Hong et al. (2023), which remains the most common approach. However, relying solely on text prompts can be insufficient for accurately capturing user intent, prompting exploration into other conditioning inputs such as static images Wu et al. (2023); Guo et al. (2024a), user sketches Zhao et al. (2023); Wang et al. (2023), human poses Zhong et al. (2024); Ma et al. (2024b); Karras et al. (2023), camera perspectives Zheng et al. (2024a); He et al. (2024a), and even additional videos Kara et al. (2024); Deng et al. (2024); Zhang et al. (2023). Given this diversity of potential conditions, unifying them into a single "any-condition" video generation framework is highly valuable.

### E.2    CONTROLLABLE VIDEO GENERATION

Recent methods like VideoComposer Wang et al. (2023), Ctrl-Adapter Lin et al. (2024b), and ControlVideo Zhao et al. (2023) have investigated any-condition video generation. Nevertheless, they still struggle with comprehensive controllability due to the complexity of multiple modalities and the limited capacity of standard diffusion or DiT encoders to interpret them. Inspired by the strong multimodal reasoning capabilities of modern MLLMs Liu et al. (2024b); Lin et al. (2024a); Wang et al. (2024a), we propose leveraging an MLLM to consolidate all possible conditions into a unified reasoning process, producing structured dense captions as inputs to a backbone Video DiT. SoTA DiT models already exhibit the capacity to interpret dense textual descriptions, as long as the input captions are sufficiently detailed in depicting both the scene and the intended generation goals. Building on this, our MLLM-based condition encoder directly addresses the comprehension bottleneck, theoretically enabling higher-quality video generation. To our knowledge, this work is the first to develop an MLLM specifically tailored for any-condition video generation. Because the caption-generation mechanism is decoupled from DiT, our proposed `Any2Caption` can be integrated into existing DiT-based methods without additional retraining.

### E.3    VIDEO CAPTIONING

Our approach is closely related to video recaptioning research, as our MLLM must produce dense captions based on the given conditions. In text-to-video settings, prior work Fan et al. (2024); Yang et al. (2024c); Nan et al. (2024); Islam et al. (2024) has demonstrated the benefits of recaptioning videos to obtain more detailed textual annotations, thereby improving the training of longer and higher-quality video generation via DiT. ShareGPT4Video Chen et al. (2024a), for example, employs GPT-4V Gpt (2023) to reinterpret video content and produce richer captions. MiraData Ju

| Type | #Inst. | #Condition | Short Caption (#Avg. Len.) | #Structured Caption(#Avg. Len.) |
|------|--------|------------|----------------------------|----------------------------------|
| Identities | 200 | 350 | 65.28 | 284.97 |
| Camera | 200 | 200 | 50.25 | 208.01 |
| Depth | 200 | 200 | 54.22 | 225.09 |
| Human Pose | 200 | 200 | 58.38 | 259.03 |
| Camera+Identities | 200 | 622 | 53.41 | 209.17 |
| Camera+Depth | 200 | 400 | 51.43 | 208.81 |
| Identities+Depth | 200 | 555 | 53.14 | 286.83 |
| Camera+Identities+Depth | 200 | 756 | 58.35 | 289.21 |

Table 7: Statistics of the constructed test datasets. **#Inst.** denotes the number of instances, and **#Condi.** indicates the number of unique conditions. **Short Cap. #Avg. Len** represents the average caption length of short captions, and **Structured Cap. #Avg. Len.** represents the average caption length of structured captions.

et al. (2024) introduces structured dense recaptioning, while InstanceCap Fan et al. (2024) focuses on instance-consistent dense recaptioning. Although we also pursue structured dense captions to enhance generation quality, our method diverges fundamentally from these previous approaches. First, because DiT models are already sufficiently powerful, we directly adopt an off-the-shelf Video DiT without incurring the substantial cost of training it with dense captions. Instead, we train an MLLM as an any-condition encoder at a comparatively lower cost; in the text-to-video scenario, for instance, we only need to train on pairs of short and dense captions, which are far easier and more abundant to obtain. Second, prior methods that recapturing the entire video risk introducing noise or even hallucinated content due to the current limitations of MLLMs in video understanding, potentially undermining DiT training quality, whereas our framework avoids this issue. Most importantly, while these approaches may rely on dense-caption-trained DiT models, real-world inference often involves very concise user prompts, creating a mismatch that can diminish final generation quality.

### E.4 MULTIMODAL LARGE LANGUAGE MODELS

Recent advances in Large Language Models (LLMs) Yang et al. (2024a) have catalyzed a surge of interest in extending their capabilities to multimodal domains Lin et al. (2024a); Li et al. (2024a); Zhang et al. (2024b). A number of works integrate a vision encoder (e.g., CLIP Liu et al. (2024b), DINOv2Tong et al. (2024), OpenCLIPLi et al. (2024b)) with an LLM, often through a lightweight "connector" module (e.g., MLP Liu et al. (2024b), Q-former Li et al. (2023)), enabling the model to process image and video inputs with minimal additional training data and parameters Liu et al. (2024b); Wu et al. (2024); Lin et al. (2024a). These approaches have demonstrated promising performance on tasks such as image captioning, visual question answering, and video understanding. Beyond purely visual data, some researchers have investigated broader modalities, such as 3D motion Chen et al. (2024b); Jiang et al. (2023) or audio Chu et al. (2024); Wu et al. (2024), thereby expanding the application range of multimodal LLMs. Despite these advances, most existing MLLMs focus on a limited set of visual modalities and do not readily accommodate more specialized inputs like human pose or camera motion. This gap restricts their ability to handle more diverse and complex conditions in the controllable video generation field. In contrast, our work targets a broader spectrum of modalities, aiming to build a single model capable of interpreting and unifying image, video, human pose, and camera conditions. Specifically, we augment an existing MLLM with dedicated encoders for motion and camera features, thereby equipping the model to process arbitrary multimodal conditions and facilitate controllable video generation.

## F EXTENDED DATASET DETAILS

**Visualization of Short and Structured Caption.** Here, we visualize the short and structured caption in Fig. 7, 8. Notably, the structured caption captures the video content in greater detail and across multiple aspects. In contrast, the short caption deliberately omits any information already supplied by a non-text condition—for example, camera movement is excluded in Fig. 7's short caption but included in Fig. 8 because it is not specified by the provided multiple-identity images. Moreover, we visualize the word distribution of the structured captions in `Any2CapIns` in Fig. 9.

Short Caption + Camera ⟶ Video

> **Short Caption:** A cozy and well-lit home. Start by showing a dining table with chairs and a chandelier, then capture the living room with a sofa and fireplace. Move towards the large windows to reveal the deck outside and the grassy area. Emphasize the warm and inviting atmosphere.

> **Structured Caption:**

1. Dense caption: An elegantly composed interior scene unfolds, where the dining and living areas are interwoven through warm tones and architectural cohesion. The narrative begins with a glimpse of a chandelier-lit dining table, then flows organically toward a plush seating arrangement and a glowing fireplace. A slow and smooth camera shift unveils tall glass windows that eventually open up to a serene deck, gracefully bridging interior comfort with exterior peace.
2. Main object caption: Key interior elements include a traditional dining ensemble centered under a sparkling chandelier, and a living room anchored by a soft sofa and an old-fashioned fireplace—framed by panoramic glass windows that serve as portals to nature.
3. Background caption: The setting is enriched by a timeless ambiance: gentle upholstery, a vintage-style wall clock, and natural daylight pouring through large windows. Outside, a sunlit wooden patio bordered by greenery extends the space beyond the glass.
4. Camera caption: Captured with a steady cam at eye-level. Begins with a lateral sweep from the dining corner before proceeding with a slow dolly movement toward the windowed wall, culminating in a reveal of the outdoor patio.
5. Style caption: Cozy and classic, with a warm color palette, soft lighting gradients, and a slightly rustic design aesthetic. The stylistic cohesion evokes a calm and welcoming atmosphere.
6. Action caption: Executing a deliberate leftward pan, the camera surveys the interior before advancing forward, transitioning the viewer's perspective from indoor warmth to the outdoor environment.

Figure 7: Illustrations of constructed short and structured captions under the camera-to-video generation.

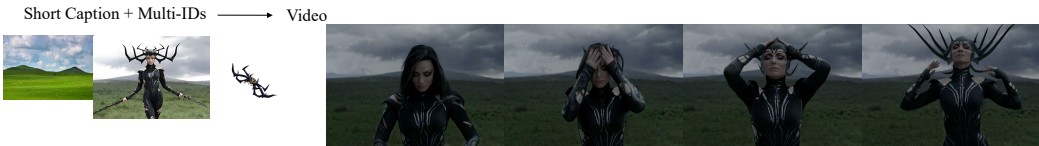

Short Caption + Multi-IDs ⟶ Video

> **Short Caption:** A person stands in a vast field under a stormy sky. They raise their hands to their face, then above their head, transforming their headpiece into a horn-like structure. The camera moves backward, capturing the gloomy atmosphere and their confident stance.

> **Structured Caption:**

1. Dense caption: Amid a thunderous, cloud-laden sky, a lone figure clad in futuristic, obsidian-colored armor stands motionless at the center of an expansive, empty field. The moment intensifies as the individual lifts their arms toward their face in a calculated motion. Gradually, their helmet morphs into an imposing, horned structure. As this metamorphosis completes, the figure returns to a commanding stance, now equipped with two elegant, blade-like weapons—one held firmly in each hand.
2. Main object caption: The armored character—fully suited in sleek, dark tech-gear—occupies the center of the frame. Their headgear shifts shape, evolving into horn-like extensions as they raise their hands in a slow, deliberate movement.
3. Background caption: Sweeping grasslands stretch beneath a foreboding sky, with rolling hills far in the distance. The atmosphere is heavy with tension, and the setting remains still, emphasizing the transformation's magnitude and isolating the character in a vast, ominous world.
4. Camera caption: Camera begins with a mid-range, centered composition. As the sequence unfolds, it subtly dollies backward to reveal more of the character's form while maintaining a consistent eye-level perspective.
5. Style caption: Dramatic and futuristic, with a moody color palette dominated by greys and blacks. The visual tone draws from dystopian cinema, focusing on solitude, metamorphosis, and subtle power.
6. Action caption: A solitary warrior lifts her arms toward her helmet, triggering its transformation. Once reformed into a horned shape, she lowers her arms, retrieves dual blades, and assumes a poised, forward-facing stance.

Figure 8: Illustrations of constructed short and structured captions under the multiIDs-to-video generation.

**Prompt Visualization for Short Caption Construction.** In Tab. 8 and 9, we show the system prompts used by GPT-4V to generate short captions. The prompt explicitly instructs GPT-4V to consider the given conditions comprehensively and produce short prompts that focus on information not covered by the non-textual inputs. For instance, when multiple identities are specified, the short prompt should avoid repeating their appearance attributes and instead highlight interactions among the identities. Conversely, when depth is the input condition, the short prompt should include more detailed appearance-related information.

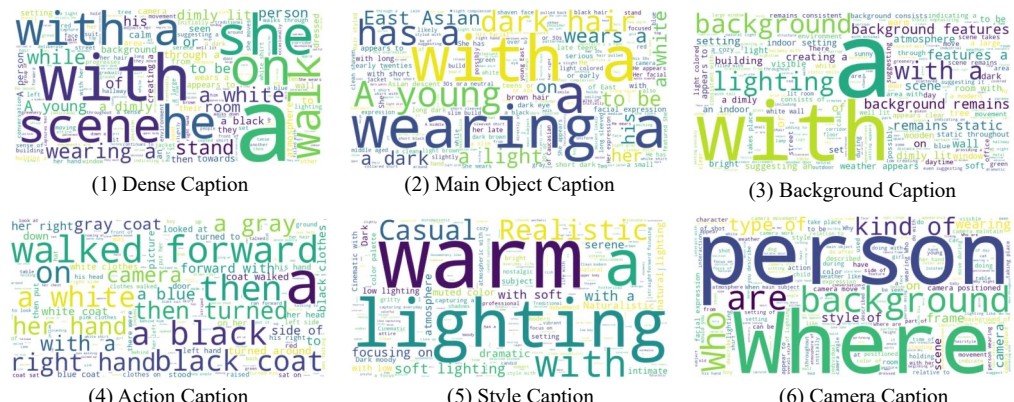

Figure 9: Word cloud of different structured captions in Any2CapIns dataset, showing the diversity.

| **Multi-IDs** | Here is the scenario: We have an MLLM model that supports a text image-conditioned intrinsic-video-caption generation task. The system input consists of:
1. A reference image composed of 2-3 horizontally stitched images (provided by the user), each stitched image containing one or several target objects for reference); and
2. A concise textual prompt (referred to as text B, the user's instruction).
The model's output is a detailed descriptive caption (**text A**) that thoroughly describes the video corresponding to the user's input prompt (**text B**) in great detail. Your task is to perform a reverse engineering. Based on the given reference image (the target objects) and the detailed target video caption (text A), you need to generate a **reasonable and concise user prompt (text B)** through your understanding, analysis, and imagination. You must adhere to the following rules:
1. Text A is a dense caption of a video, including all the key objects, their attributes, relationships, background, camera movements, style, and more. Carefully analyze this caption for all relevant details.
2. Analyze the provided reference images in detail to identify the differences or missing details compared to the target video description. These may include environment details, the interaction between objects, the progression of actions, camera movements, style, or any elements not covered by the reference image. Based on these analyses, generate the user's instructions.
3. The user's prompt must include the following aspects: first, an overall description of where the target objects are and what they are doing, along with the temporal progression of their actions. Then, it should describe the background, style, and camera movements.
4. If the target video introduces new objects not present in the reference images, the user's prompt should describe the attributes of the new target objects and their interactions with the other target objects.
5. If the video's style differs from the reference, briefly describe the style in a few words.
6. When the background needs to be described, include details about people, settings, and styles present in the background.
7. Avoid repeating information that can be inferred from the reference images, and eliminate redundant descriptions in the user prompt.
8. The user prompt (text B) must be written in simple wording, maintaining a concise style with short sentences.
9. The user's instructions should vary in expression; For example, prompts do not always need to start with the main subject. They can begin with environmental details, camera movements, or other contextual aspects.
Here are three examples representing the desired pattern:
================================================================
   [In-context Examples]
================================================================
   [Input] |

Table 8: Demonstration of the prompt used for GPT-4V to generate the short prompt when the input condition is the multi-IDs.

## G  MORE STATISTICS INFORMATION OF IRSCORE

We generate a total of 15,378 question-answer (QA) pairs, averaging 19.2 pairs per structured caption. Fig. 10 presents the distribution of constructed questions across different aspects in the structured caption, and Tab. 10 shows representative QA pairs for each aspect. Notably, questions under the *main object* category emphasize fine-grained details such as clothing color or hairstyles, while *action* questions focus on object interactions and movements. This level of specificity allows us to rigorously assess whether the generated captions are both complete and precise.

**Depth** Here is the scenario: We have an MLLM model that supports a text & image-conditioned intrinsic-video-caption generation task. The system input consists of:

1. A reference image composed of 3-5 horizontally stitched depth maps in temporal sequence (provided by the user, each map containing depth information for reference); and

2. A concise textual prompt (referred to as text B, the user's instruction).

The model's output is a detailed descriptive caption (text A) that thoroughly describes the video corresponding to the user's input prompt (text B) in great detail. Now, I need you to perform a reverse engineering task. Based on the given reference image (the depths) and the detailed target video caption (text A), you must generate a reasonable and concise user prompt (text B) through your understanding, analysis, and imagination. To ensure accurate and effective outputs, follow these rules strictly:

1. Text A is a dense caption of a video, including all the key objects, their attributes, relationships, background, camera movements, style, and more. Carefully analyze this caption to extract the necessary details.

2. Since the depth information already provides the necessary geometric outlines and layout details. Do not repeat this information in the user prompt. Instead, focus on the aspects not covered by the depth maps.

3. The user's instruction should highlight details not included in the depth map, such as environmental details, the appearance of the subjects, interactions between subjects, the progression of actions, relationships between the subjects and the environment, camera movements, and overall style.

4. For dense depth maps (more than 5 maps), assume the maps provide the camera movements and actions between objects, focusing on describing the appearance of the subjects and environment, the atmosphere, and subtle interactions between subjects and their environment.

5. For sparse depth maps (5 maps or fewer), assume the maps only provide scene outlines. Emphasize details about the subjects' appearance, environment, interactions between subjects, relationships between subjects and the environment, and camera movements.

6. The user prompt (text B) must be written in simple wording, maintaining a concise style with short sentences, with a total word count not exceeding 100.

7. Your output should be a continuous series of sentences, not a list or bullet points.

8. The user's instructions should vary in expression; they don't always need to begin with a description of the main subject. They could also start with environmental details or camera movements.

Here are three examples representing the desired pattern:

========================================================================

[In-context Examples]

========================================================================

[Input]

Table 9: Demonstration of the prompt used for GPT-4V to generate the short prompt when the input condition is the depth.

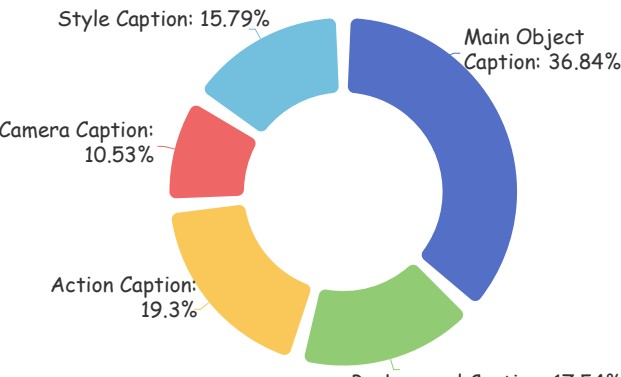

Figure 10: QA pairs proportion in structured captions.

# H   DETAILED SETUPS

## H.1   DETAILED TESTING DATASET

Here, we present the statistics of the test in Tab. 7, which covers four types of single conditions (e.g., *Depth*, *Camera*, *Identities*, and *Human pose*), and four types of compositional conditions(e.g., *Camera+Identities*, *Camera+Depth*, *Identities+Depth* and *Camera+Identities+Depth*). Each category contains 200 instances.

| Aspect | QA Pairs |
|---|---|
| Main Object | What is the young woman adjusting as she walks down the corridor? Her wide-brimmed hat. |
| | What color is the young woman's T-shirt? Light blue. |
| | How does the young woman feel as she walks down the corridor? Happy and carefree. |
| | What is the young woman wearing? Light blue t-shirt with pink lettering, blue jeans, and a wide-brimmed hat. |
| | What is the young woman's hair length? Long. |
| | What is the position of the young woman in the frame? In the center of the frame. |
| | What is the main object in the video? A large shark. |
| | What is the color of the underwater scene? Blue. |
| | What are the two scientists wearing? White lab coats and gloves. |
| | What is the first scientist using? A microscope. |
| Background | Where is the young woman walking? Down a corridor. |
| | What time of day does the scene appear to be set? Daytime. |
| | What can be seen in the background of the corridor? Beige walls and large windows. |
| | What is the weather like in the video? Clear. |
| | Where is the shark located? On the ocean floor. |
| | What surrounds the shark in the video? Smaller fish. |
| | Where is the laboratory setting? In a brightly lit environment with shelves filled with bottles. |
| | What detail does the background highlight? The scientific setting with static emphasis. |
| Camera | How does the camera follow the young woman? Moving backward |
| | What is the camera's height relative to the person? Roughly the same height as the person. |
| | What shot type does the camera maintain? Medium close-up shot of the upper body. |
| | How does the camera position itself to capture the subject? At a higher angle, shooting downward. |
| | How does the camera capture the environment? From a medium distance. |
| | How is the camera positioned? At approximately the same eye level as the subjects, maintaining a close-up shot. |
| | How does the camera move in the video? It pans to the right. |
| Style | What is the style of the video? Casual and candid. |
| | What kind of design does the corridor have? Modern and clean design. |
| | What style does the video portray? Naturalistic style with clear, vivid visuals. |
| | What does the video style emphasize? Clinical, high-tech, and scientific precision. |
| | What is the color theme of the lighting? Bright and cool. |
| | What kind of atmosphere does the laboratory have? Professional and scientific. |
| Action | What does the young woman do with both hands occasionally? Adjusts her hat. |
| | What is the young woman doing as she moves? Walking forward with her hands on her hat. |
| | What is the main action of the shark in the video? Lying motionless. |
| | What is the movement of the fish like? Calm and occasionally darting. |
| | What is the movement of the first scientist at the beginning? Examines a microscope. |
| | What task is the second scientist engaged in? Handling a pipette and a beaker filled with green liquid. |
| | How does the second scientist transfer the liquid? Carefully using a pipette into the beaker. |
| | Are there any noticeable movements in the background? Occasional small particles floating. |

Table 10: Demonstration of generated question-answer pairs utilized in IRSCORE calculation.

| Configuration | Stage-1: Alignment Learning | | Stage-2: Condition-Interpreting Learning | | | |
|---|---|---|---|---|---|---|
| | Camera | Motion | Identities | Human pose | Camera | Depth |
| Optimizer | AdamW | AdamW | AdamW | AdamW | AdamW | AdamW |
| Precision | bfloat16 | bfloat16 | bfloat16 | bfloat16 | bfloat16 | bfloat16 |
| Learning Rate | 5e5 | 5e5 | 5e5 | 5e5 | 5e5 | 1e5 |
| Weight Decay | 0.01 | 0.01 | 0.01 | 0.01 | 0.01 | 0.01 |
| Joint Train Ratio | 0.0 | 0.0 | 0.0 | 0.4 | 0.6 | 0.8 |
| Betas | (0.9, 0.99) | (0.9, 0.99) | (0.9, 0.99) | (0.9, 0.99) | (0.9, 0.99) | |
| Dropout Possibility | 0.0 | 0.0 | 0.4 | 0.6 | 0.6 | 0.6 |
| Dropout (Short Cap.) | 0.0 | 0.0 | 0.6 | 0.6 | 0.6 | 0.6 |
| Batch Size Per GPU | 4 | 4 | 4 | 4 | 4 | 4 |
| Training Data | Camera Movement Description Dataset (Manual) | Action Description Dataset (Manual) | MultiIDs | Human Pose LLaVA-150K | Camera LLaVA-150K | Depth LLaVA-150K Alpaca-50K |

Table 11: Training recipes for `Any2Caption`.

## H.2 IMPLEMENTATION DETAILS

We leverage Qwen2VL-7B Wang et al. (2024a) as the backbone of our model, which supports both image and video understanding. The human pose in the input conditions is encoded and processed

in the video format. The camera encoder adopts the vision encoder architecture, with the following settings: in channels set to 96, patch size of 16, depth of 8, and 8 attention heads. During training, to simulate the brevity and randomness of user inputs, we randomly drop sentences from the short caption with a dropout rate of 0.6; a similar dropout strategy is applied to non-textual conditions. We conducted the training on 8×A800 GPUs.

### H.3 Detailed Training Procedure

We employ a two-stage training process to enhance the alignment and interpretability of multimodal conditions in `Any2Caption`.

**Stage-1: Alignment learning.** This stage focuses on aligning features extracted by the camera encoder with the LLM feature space. To achieve this, we first extract camera movement descriptions (e.g., *fixed*, *backward*, *pan to the right*) from the camera captions in `Any2CapIns` to construct a camera movement description dataset. We then introduce two specialized tokens, `<|camera_start|>` and `<|camera_end|>`, at the beginning and end of the camera feature embeddings. During training, only the camera encoder is optimized, while all other parameters in `Any2Caption` remain frozen. Similarly, for motion alignment, we construct a motion description dataset by extracting action descriptions (e.g., *walking*, *dancing*, *holding*) from the action captions in `Any2CapIns`. We then freeze all model parameters except those in the motion encoder to ensure the LLM effectively understands motion-related conditions.

**Stage-2: Condition-Interpreting Learning.** After alignment learning, we initialize `Any2Caption` with the pre-trained Qwen2-VL, motion encoder, and camera encoder weights. We then employ a progressive mixed training strategy, updating only the `lm_head` while keeping the multimodal encoders frozen. The training follows a sequential order based on condition complexity: identities ⇒ human pose ⇒ camera ⇒ depth. Correspondingly, the integration ratio of additional vision/text instruction datasets is progressively increased, set at 0.0, 0.4, 0.6, and 0.8, ensuring a balanced learning process between condition-specific specialization and generalization.

### H.4 Detailed Implementations

In Tab. 11, we list the detailed hyperparameter settings in two stages. All the training is conducted on 8×A800 (80G) GPUs.

## I Extended Experiment Results and Analyses

### I.1 The Effectiveness of the AnyCaption.

We evaluate the performance of our proposed model, AnyCaption, against various baselines on the detailed video captioning benchmark VDC Chai et al. (2025). This evaluation is designed to inversely assess whether the model can accurately generate fine-grained content aligned with the target video. As shown in Table 12, our model consistently outperforms all baseline video-language models across multiple categories. In particular, it achieves the best results in detail-sensitive aspects such as camera perspective and main object descriptions. These results highlight the strong capability of AnyCaption in capturing fine-grained visual semantics and generating detailed, faithful video descriptions which are an essential prerequisite for high-fidelity video generation. Furthermore, we conduct additional comparisons on our proposed benchmark, with results presented in Table 13. AnyCaption also outperforms all baselines across all evaluated dimensions, demonstrating its effectiveness in capturing users' core intentions and generating detailed, goal-consistent descriptions of target videos.

### I.2 The Capability for Understanding Complex Instruction.

We further examine `Any2Caption`'s ability to handle complex user instructions, particularly regarding whether it accurately captures the user's intended generation targets. From Fig. 11, we observe that the model focuses precisely on the user-specified main objects, such as a "woman warrior" or a background "filled with chaos and destruction"—when producing structured captions.

| Model | Camera | | Short | | Background | | Main Object | | Detailed | | Overall | |
|---|---|---|---|---|---|---|---|---|---|---|---|---|
| | Acc | Score | Acc | Score | Acc | Score | Acc | Score | Acc | Score | Acc | Score |
| LLaMA-VID Li et al. (2024a) | 39.47 | 2.10 | 29.92 | 1.56 | 38.01 | 1.45 | 31.24 | 1.59 | 25.67 | 1.38 | 32.86 | 1.62 |
| Video-ChatGPT-7B Maaz et al. (2024) | 37.46 | 2.00 | 28.92 | 1.51 | 35.89 | 1.37 | 30.13 | 1.50 | 25.02 | 1.34 | 31.48 | 1.54 |
| Video-LLaVA-7B Lin et al. (2024a) | 37.48 | 1.97 | 30.74 | 1.58 | 35.80 | 1.38 | 30.19 | 1.46 | 25.22 | 1.33 | 31.89 | 1.54 |
| LongVA-7B Zhang et al. (2024a) | 35.32 | 1.94 | 29.35 | 1.53 | 33.91 | 1.33 | 30.34 | 1.49 | 25.40 | 1.33 | 30.86 | 1.52 |
| LLaVA-NeXT-V7B Zhang et al. (2024d) | 36.93 | 1.96 | 30.59 | 1.56 | 36.10 | 1.43 | 31.79 | 1.50 | 26.01 | 1.35 | 32.28 | 1.56 |
| LLaVA-1.6-7B Liu et al. (2024c) | 36.56 | 1.91 | 30.94 | 1.59 | 35.62 | 1.38 | 32.06 | 1.51 | 26.31 | 1.34 | 32.29 | 1.55 |
| ShareGPTVideo-8B Chen et al. (2024a) | 39.16 | 2.06 | **32.61** | **1.70** | **38.90** | **1.99** | 34.89 | 1.68 | 29.81 | 1.52 | 35.07 | 1.79 |
| LLaVA-OV-7B Li et al. (2025) | 39.02 | 2.04 | 31.84 | 1.64 | 36.52 | 1.84 | 33.24 | 1.65 | 28.41 | 1.48 | 33.81 | 1.73 |
| Qwen2-VL-7B Wang et al. (2024a) | 38.25 | 1.98 | 30.58 | 1.61 | 36.79 | 1.86 | 37.49 | 1.96 | 32.23 | 1.61 | 35.07 | 1.80 |
| Any2Caption | **40.26** | **2.19** | 31.88 | 1.64 | 38.80 | 1.96 | **41.50** | **2.06** | 35.13 | 1.87 | 37.51 | 1.94 |

Table 12: Quantitative comparison on the VDC video captioning benchmark Chai et al. (2025). We report VDCscore across five caption aspects: camera captions, short captions, background captions, main object captions, and detailed captions. The final two columns show the overall performance.

| Model | Dense | | Main Object | | Background | | Action | | Style | | Camera | |
|---|---|---|---|---|---|---|---|---|---|---|---|---|
| | Acc | Qual. | Acc | Qual. | Acc | Qual. | Acc | Qual. | Acc | Qual. | Acc | Qual. |
| LLaVA-NeXT-V7B Zhang et al. (2024d) | 65.23 | 2.81 | 43.57 | 2.13 | 55.41 | 2.16 | 40.87 | 1.72 | 52.64 | 2.43 | 35.97 | 1.90 |
| LLaVA-1.6-7B Liu et al. (2024c) | 66.01 | 2.83 | 44.02 | 2.17 | 54.98 | 2.14 | 39.11 | 1.75 | 52.91 | 2.41 | 36.13 | 1.92 |
| ShareGPTVideo-8B Chen et al. (2024a) | 75.36 | 3.35 | 54.90 | 2.68 | 66.77 | 2.65 | 51.92 | 2.05 | 62.48 | 2.94 | 55.29 | 2.68 |
| Qwen2-VL-7B Wang et al. (2024a) | 71.42 | 3.14 | 51.31 | 2.41 | 63.51 | 2.45 | 50.77 | 1.93 | 59.18 | 2.75 | 51.22 | 2.41 |
| Any2Caption | **79.45** | **3.81** | **56.29** | **2.78** | **70.07** | **2.71** | **56.78** | **2.14** | **65.74** | **3.15** | **67.25** | **3.79** |

Table 13: Comparison between Video-LLMs baselines and Any2Caption on our proposed benchmark. Since the baseline models are limited to handling only video and image inputs, we conduct experiments on the *Depth* and *Multiple Identities* benchmarks. For a fair comparison, we adopt a one-shot setting for the baseline models, enabling them to generate detailed descriptions and structured outputs for the target videos. We report the average scores of intention reasoning accuracy (Acc) and quality (Qual.) across all tasks.

In contrast, a short caption combined with condition captions often includes extraneous objects or background details present in the identity images, which distract from the user's intended targets in the final video generation.

Additionally, we assess the model's performance on instructions containing implicit objects or actions, as shown in Fig. 12 and 13. In these examples, the model correctly interprets phrases like "the most right person" as "a young Black woman with long, curly brown hair, wearing a black and white outfit" and similarly associates implicitly specified objects with the provided conditions, generating structured captions that align with the user's goals.

Lastly, Fig. 16 compares videos generated using different captions. The results indicate that structured captions significantly improve both the smoothness of motion and the overall consistency of the generated videos.

### I.3    MORE VIDEO VISUALIZATION

Here, we compare the results of different customized models Ju et al. (2025) after integrating structured captions. Fig. 14 shows camera-controlled video generation results, and Fig. 15 illustrates depth-controlled outcomes. We observe that structured captions improve image quality and motion smoothness by providing richer scene details.

For multi-identity-controlled video generation, as depicted in Fig. 16, 17, and 18, structured captions lead to more expressive and realistic representations of the referenced identities, with more accurate colors and smoother actions.

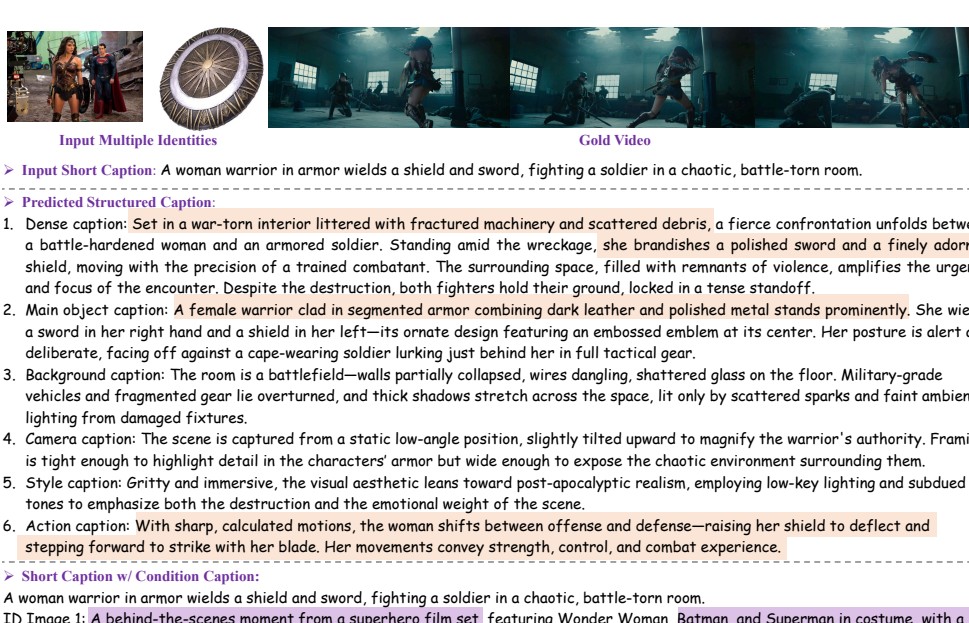

**Input Multiple Identities**  **Gold Video**

➢ **Input Short Caption:** A woman warrior in armor wields a shield and sword, fighting a soldier in a chaotic, battle-torn room.

-----------------------------------------------------------------------------

➢ **Predicted Structured Caption:**

1. Dense caption: Set in a war-torn interior littered with fractured machinery and scattered debris, a fierce confrontation unfolds between a battle-hardened woman and an armored soldier. Standing amid the wreckage, she brandishes a polished sword and a finely adorned shield, moving with the precision of a trained combatant. The surrounding space, filled with remnants of violence, amplifies the urgency and focus of the encounter. Despite the destruction, both fighters hold their ground, locked in a tense standoff.

2. Main object caption: A female warrior clad in segmented armor combining dark leather and polished metal stands prominently. She wields a sword in her right hand and a shield in her left—its ornate design featuring an embossed emblem at its center. Her posture is alert and deliberate, facing off against a cape-wearing soldier lurking just behind her in full tactical gear.

3. Background caption: The room is a battlefield—walls partially collapsed, wires dangling, shattered glass on the floor. Military-grade vehicles and fragmented gear lie overturned, and thick shadows stretch across the space, lit only by scattered sparks and faint ambient lighting from damaged fixtures.

4. Camera caption: The scene is captured from a static low-angle position, slightly tilted upward to magnify the warrior's authority. Framing is tight enough to highlight detail in the characters' armor but wide enough to expose the chaotic environment surrounding them.

5. Style caption: Gritty and immersive, the visual aesthetic leans toward post-apocalyptic realism, employing low-key lighting and subdued tones to emphasize both the destruction and the emotional weight of the scene.

6. Action caption: With sharp, calculated motions, the woman shifts between offense and defense—raising her shield to deflect and stepping forward to strike with her blade. Her movements convey strength, control, and combat experience.

-----------------------------------------------------------------------------

➢ **Short Caption w/ Condition Caption:**

A woman warrior in armor wields a shield and sword, fighting a soldier in a chaotic, battle-torn room.

ID Image 1: A behind-the-scenes moment from a superhero film set, featuring Wonder Woman, Batman, and Superman in costume, with a film crew and green screen in the background.    *Introduce noise information*

ID Image 2: A round shield with intricate designs and a central starburst pattern.

Figure 11: Illustrations of input short caption, predicted structured caption, and short caption combined with condition captions. The condition captions may introduce noisy information due to the focus distraction of the background of input identities, for example, the *Batman*, *Superman*, *film crew*.

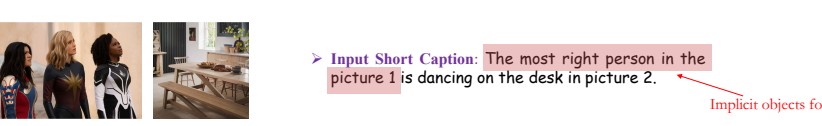

**Input Multiple Identities**

➢ **Input Short Caption**: The most right person in the picture 1 is dancing on the desk in picture 2.    *Implicit objects for generation*

-----------------------------------------------------------------------------

➢ **Predicted Structured Caption:**

1. Dense caption: Inside a sunlit modern kitchen, a young woman confidently dances atop a wooden dining table, her movements vibrant and self-assured. Clad in a monochrome outfit that blends fashion with flair, she glides, twirls, and gestures to an internal rhythm. Light pours in from generous windows, casting soft highlights on the table's surface—where everyday elements like fruit and a meal serve as subtle reminders of domestic life. The warm tones of the wood and the greenery visible outside enhance the feeling of life, spontaneity, and celebration.

2. Main object caption: A confident Black woman with voluminous, curly brown hair performs expressive dance movements on a dining table. She wears a black-and-white color-block jacket, slim black pants, and ankle boots. Every motion of her body radiates joy, from the sway of her arms to the rhythm in her steps, all anchored at the center of the scene.

3. Background captionThe room is bright and contemporary—featuring polished wooden cabinets, clean surfaces, and a dining set casually set with fruit and a dish. Beyond the expansive windows lies a view of lush foliage, infusing the interior with calm, natural energy. Nothing in the setting feels staged—it's a real kitchen full of warmth and life.

4. Camera caption: The scene is observed from a slightly elevated camera angle, keeping the subject fully in frame while subtly emphasizing the table's height and the dancer's grounded energy.

5. Style caption: Effortless and candid. The style emphasizes natural light, domestic textures, and a relaxed color palette, evoking authenticity and comfort with a hint of artistic spontaneity.

6. Action caption: She steps, spins, and gestures with increasing enthusiasm, transforming the tabletop into her stage. Her expressive movements seem improvised yet intentional, turning an everyday moment into a spontaneous performance.

Figure 12: Illustrations of predicted structured captions based on the input multiple identities and the short instruction that expresses the implicit objects for the video generation.

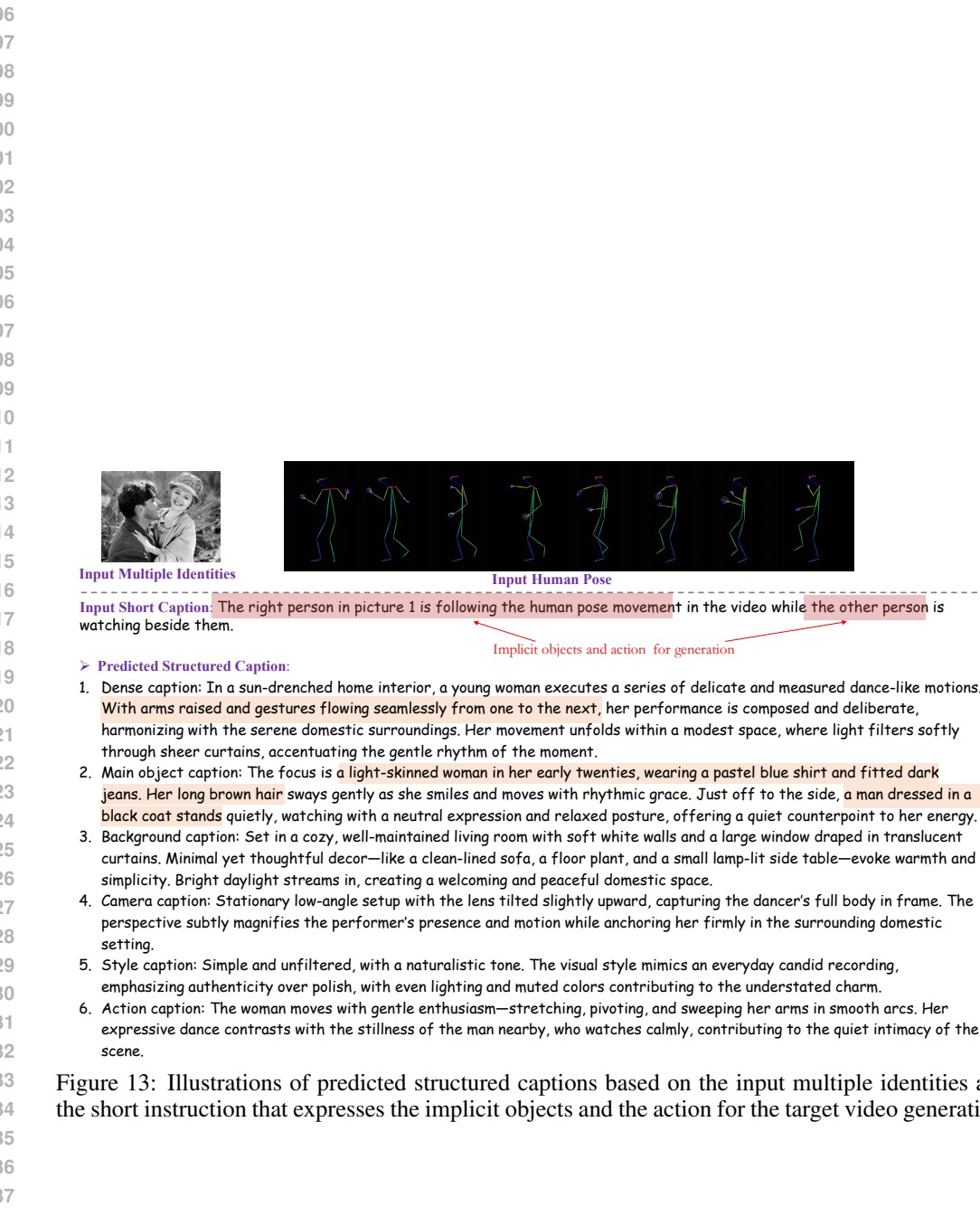

**Input Multiple Identities**

**Input Human Pose**

**Input Short Caption:** The right person in picture 1 is following the human pose movement in the video while the other person is watching beside them.

*Implicit objects and action for generation*

➢ **Predicted Structured Caption:**

1. Dense caption: In a sun-drenched home interior, a young woman executes a series of delicate and measured dance-like motions. With arms raised and gestures flowing seamlessly from one to the next, her performance is composed and deliberate, harmonizing with the serene domestic surroundings. Her movement unfolds within a modest space, where light filters softly through sheer curtains, accentuating the gentle rhythm of the moment.

2. Main object caption: The focus is a light-skinned woman in her early twenties, wearing a pastel blue shirt and fitted dark jeans. Her long brown hair sways gently as she smiles and moves with rhythmic grace. Just off to the side, a man dressed in a black coat stands quietly, watching with a neutral expression and relaxed posture, offering a quiet counterpoint to her energy.

3. Background caption: Set in a cozy, well-maintained living room with soft white walls and a large window draped in translucent curtains. Minimal yet thoughtful decor—like a clean-lined sofa, a floor plant, and a small lamp-lit side table—evoke warmth and simplicity. Bright daylight streams in, creating a welcoming and peaceful domestic space.

4. Camera caption: Stationary low-angle setup with the lens tilted slightly upward, capturing the dancer's full body in frame. The perspective subtly magnifies the performer's presence and motion while anchoring her firmly in the surrounding domestic setting.

5. Style caption: Simple and unfiltered, with a naturalistic tone. The visual style mimics an everyday candid recording, emphasizing authenticity over polish, with even lighting and muted colors contributing to the understated charm.

6. Action caption: The woman moves with gentle enthusiasm—stretching, pivoting, and sweeping her arms in smooth arcs. Her expressive dance contrasts with the stillness of the man nearby, who watches calmly, contributing to the quiet intimacy of the scene.

Figure 13: Illustrations of predicted structured captions based on the input multiple identities and the short instruction that expresses the implicit objects and the action for the target video generation.

➢ **Short Caption:** A serene video of a large house with a red roof and a spacious porch, surrounded by lush greenery. A peaceful countryside setting with vibrant colors and a tranquil atmosphere.

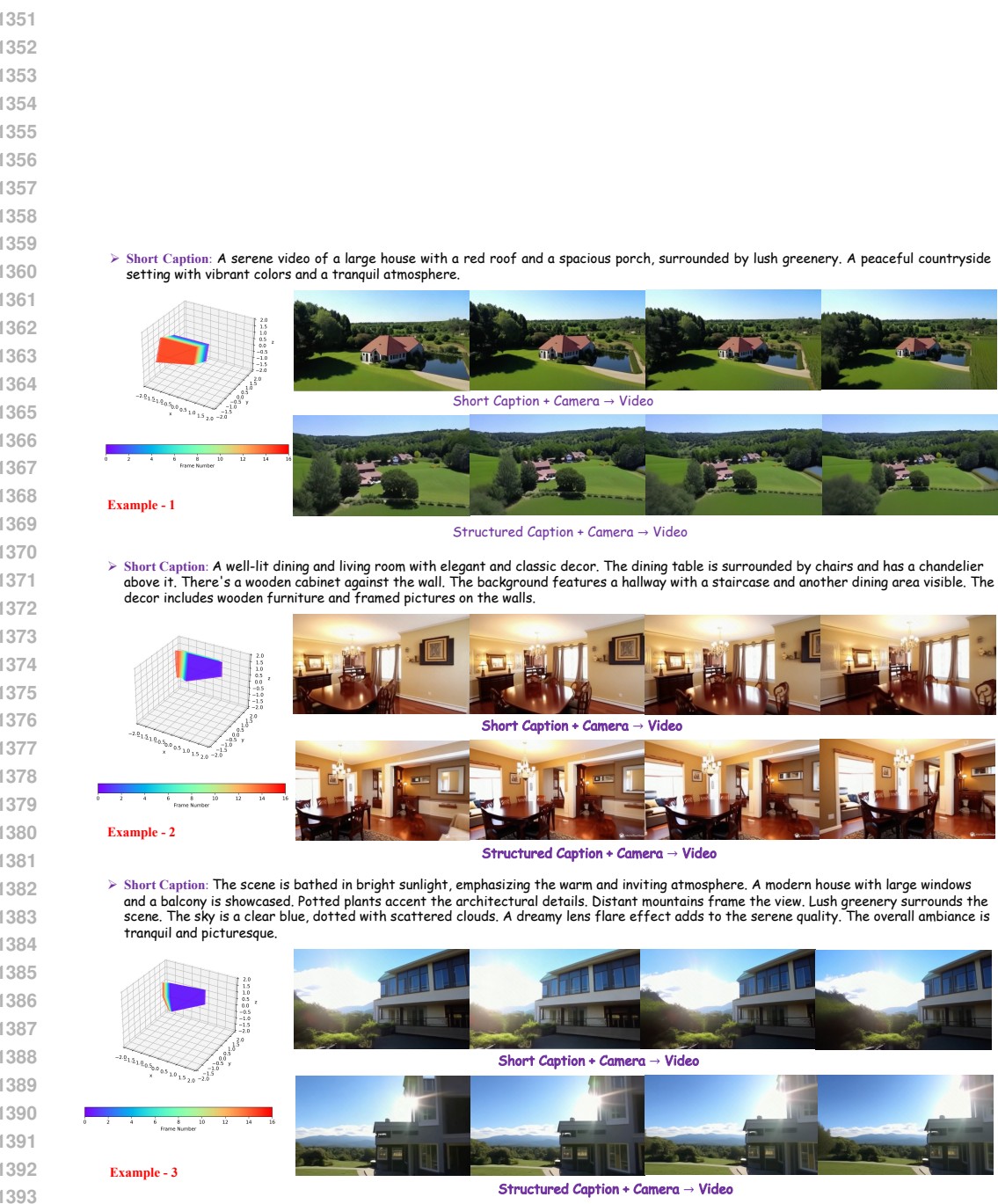

Figure 14: Illustrations of predicted structured captions based on the input multiple identities and the short instruction that expresses the implicit objects and the action for the target video generation.

➢ **Short Caption**: A dark, static background enhances the brightly colored, rotating spiral of small blocks. The camera remains fixed, capturing the mesmerizing effect of the colors shifting subtly. The dynamic movement creates a hypnotic, abstract visual style.

**Example – 4**

**Short Caption + Depth → Video**

**Structured Caption + Depth → Video**

➢ **Short Caption**: The park is sunny and lush with green trees and a small pond. A young couple in their late twenties embraces and shares a kiss. The woman, in a white sleeveless wedding dress and holding a bouquet, playfully touches the man's face. He is dressed in a black suit. The scene is romantic and intimate, with soft, natural lighting. The camera pans gently, capturing their affectionate interaction and the serene environment.

**Example – 5**

**Short Caption + Depth → Video**

**Structured Caption + Depth → Video**

Figure 15: Illustrations of predicted structured captions based on the input multiple identities and the short instruction that expresses the implicit objects and the action for the target video generation.

➤ **Short Caption:** A woman in a military uniform talks on the phone while holding a document, standing beside a man in a blue uniform against a wall. The setting is formal and professional, suggesting an official procedure in an institutional environment with light-colored walls and framed documents. The camera captures their upper bodies, moving backward and tilting upward, transitioning from a close-up to a medium close-up shot.

**Short Caption + Identities → Video**

**Short Caption w/ Condition Caption + Identities → Video**

**Example - 6**

**Structured Caption + Identities → Video**

➤ **Short Caption:** A young woman in a traditional colorful outfit rides a galloping black horse through a lush green landscape. The camera follows her movements, capturing the dynamic and vibrant scene, with her hair flowing in the wind. The background is blurred to emphasize the speed and joy of the rider. The overall feel is natural and bright.

**Short Caption + Identities → Video**

**Short Caption w/ Condition Caption + Identities → Video**

**Example -7**

**Structured Caption + Identities → Video**

Figure 16: Illustrations of predicted structured captions based on the input multiple identities and the short instruction that expresses the implicit objects and the action for the target video generation.

➤ **Short Caption:** A martial arts dojo scene where an instructor in black demonstrates techniques, throwing a student in white to the ground. Students sit in a circle on the green mat floor, observing attentively. In the background, banners and signs indicate martial arts training, with a seated audience and standing spectators. The camera moves from a medium to a long shot, capturing the full scene with respect and focus.

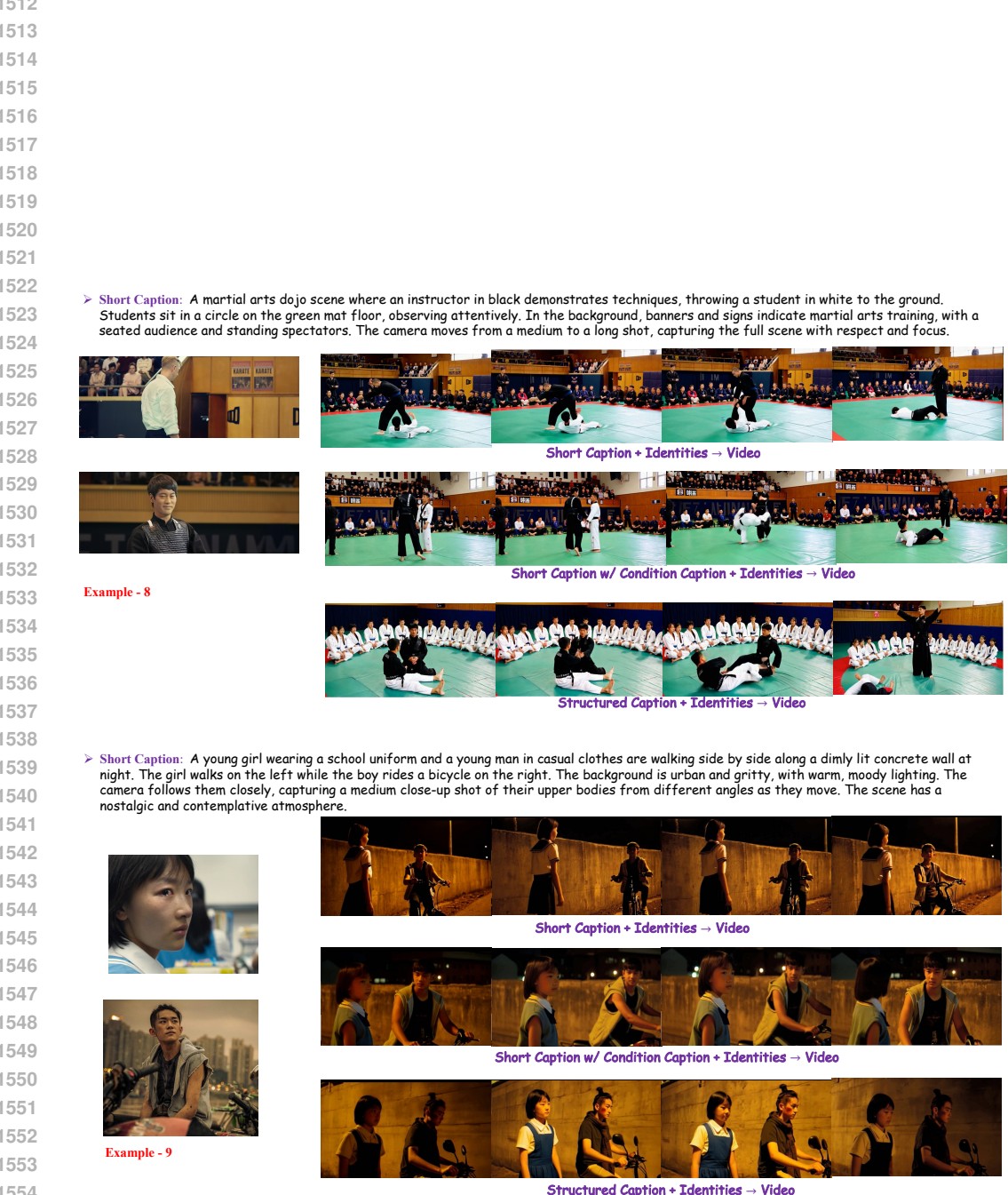

➤ **Short Caption:** A young girl wearing a school uniform and a young man in casual clothes are walking side by side along a dimly lit concrete wall at night. The girl walks on the left while the boy rides a bicycle on the right. The background is urban and gritty, with warm, moody lighting. The camera follows them closely, capturing a medium close-up shot of their upper bodies from different angles as they move. The scene has a nostalgic and contemplative atmosphere.

Figure 17: Illustrations of predicted structured captions based on the input multiple identities and the short instruction that expresses the implicit objects and the action for the target video generation.

> **Short Caption**: A woman wearing fashionable clothes stands in the room, smiling and showing the goods in her hand. Then the camera zooms in and focuses on the details of the goods in the person's hand.

Short Caption + Identities → Video

Short Caption w/ Condition Caption + Identities → Video

**Example - 10**

Structured Caption + Identities → Video

> **Short Caption**: Two cartoon characters are smiling at the camera together.

Short Caption + Identities → Video

Short Caption w/ Condition Caption + Identities → Video

**Example - 11**

Structured Caption + Identities → Video

Figure 18: Illustrations of predicted structured captions based on the input multiple identities and the short instruction that expresses the implicit objects and the action for the target video generation.

