# OpenReview forum: "Interpreting Any Condition to Caption for Controllable Video Generation"
_ICLR.cc/2026/Conference — ICLR 2026 Conference Withdrawn Submission_

### Official Review · Reviewer_rStu · 2025-10-28

**Soundness:** 2
**Presentation:** 1
**Contribution:** 2
**Rating:** 2
**Confidence:** 4

**Summary:**

The paper presents a pipeline to generate structured text prompts based on various input conditioning signals (text, human pose, depth, segmentation, camera trajectory, etc) for text-conditional video generation. For this, the authors curate a dataset with various synthetic annotations. Then, synthetic structured text prompts are generated for it using GPT-4V and LLaVA. Then, they are manually filtered. Then, they are converted to short captions with GPT-4V. Then, the authors train (fine-tune from existing components) an MLLM to generate long captions based on short captions and annotations. This MLLM is then used at inference time to generate structured captions for video models (MotionCtrl, CameraCtrl, ControlVideo, etc.). The authors test influence of structured captions for several existing video generators and demonstrate improvement.

**Strengths:**

- The paper presents a model which is able to do prompt enhancement using the side conditioning information (depth/sketch/segmentation/etc). This can help mitigate contradictions between enhanced captions and the corresponding side annotations.
- The paper presents a dataset with synthetic annotations which could be helpful for video model fine tunings in the industry (depending on the quality and the license: the paper does not disclose these details).

**Weaknesses:**

- I am convinced that this paper should be a technical report, rather than an academic submission. I do not understand what insights the paper intents to convey. The fact that a VLLM can generate structured captions from diverse inputs is obvious. The fact that structured captions improve the video generation performance is well-known.
-  It is misleading to say that the method converts any condition into a caption, because it makes a reader think that such dense conditions like depth are converted to some intricate caption and the video generator does not need to be conditioned on a depth map anymore. At least it was my original impression after reading the paper for the first ~20 minutes. Because of that, I find the "Any2Caption" pipeline to be misleading.
- The qualitative examples from the supplementary do not include the accompanying dense prompts which were used to produce them.
- It would be good to include more qualitatives
- It's unclear why the method is tested on outdated backbones (MotionCtrl/CameraCtrl, ControlVideo, etc) and why Wan2.1/Wan2.2 model is excluded from the comparisons
- It's not obvious why we need to generate long captions using the conditioning signals. I guess it is only necessary for dense conditioning like depth/sketch/human pose because otherwise long captions can contradict the conditionings. But if that's the case, it should be clearly shown. Overall, i find the vanilla prompt enhancement baseline to be missing from the comparisons.

**Questions:**

- Am I getting it right that for depth2video, the model is still conditioned on the original depth map? (e.g., for ControlVideo, the original ControlNet is still being used with the corresponding depth map input).

Typos: "how well" => "how good" (in Sec 6.2)

---

### Official Review · Reviewer_WxPH · 2025-10-30

**Soundness:** 2
**Presentation:** 3
**Contribution:** 2
**Rating:** 4
**Confidence:** 5

**Summary:**

This paper proposes Any2Caption, a framework that turns a vision–language model into a structured re-captioning model conditioned on multiple modalities (depth, pose, camera, identity, etc.). The method generates six-part structured captions (dense, object, background, action, style, camera) from arbitrary multimodal inputs, which can then be fed into existing text-to-video models to improve controllability and visual quality.

**Strengths:**

- The paper is well-written and clearly explains motivation and method.

 - The proposed structured-captioning paradigm is intuitive yet effective, successfully turning a general VLM into a condition-aware re-captioning model.

- The implementation is efficient, requiring no modification to downstream video generators.

- The results on multiple generators show that longer, structured captions can improve controllability and consistency to some extent.

**Weaknesses:**

**Distribution Shift and Potential Suboptimal Improvement**

The paper evaluates multiple video generation models in a zero-shot setting, but the structured caption model was likely never exposed to them during training.
Since different generators prefer different caption styles, the improvement may simply come from length improvement rather than genuine interpretive ability(as many video generators are trained on dense captions, and Any2Caption’s outputs are also long and verbose, which could coincidentally fit those models better).
This suggests that the observed improvement might be partly artefactual and potentially suboptimal rather than reflecting a true generalization capability.

**Limited Necessity Demonstration**

In RQ1 (“Is the structured caption necessary?”), the authors intentionally choose the multi-ID condition, which is a case where structured rewriting may help the most due to better seperation of semantic entities among input identities (though such seperation can potentially be achieved without structured caption).

This makes this ablation reasonable but also biased: it only shows necessity for this task, potentially most favorable setting.

The same necessity is not demonstrated for other settings, where structured captions may be unnecessary or even redundant.

Thus, RQ1 provides only partial evidence and does not establish the claimed general necessity of structured captioning.


**Inadequate Baselines and Limited Practical Impact**

As a paper essentially proposing a prompt enhancer, Any2Caption should be compared not only with short prompts but also with existing prompt enhancers. Most well-known open-sourced video generation models/projects already include prompt enhancers, often large LLMs fine-tuned or prompted via In-context learning to generate model-preferred prompts (e.g., CogVideoX’, Hunyuan’s, etc.). Additionally, as author mentioned, many video recaptioning methods are also proposed to enhance video generation ( ShareGPT4Video, InstanceCap, etc.)
Without such baselines, it remains unclear whether Any2Caption provides benefits over existing prompt optimization strategies.

From a more practical standpoint, for instance, if a user is already employing models such as HunyuanVideo or CogVideoX, both of which feature built-in prompt enhancers optimized for their respective training data, it is not obvious why one would replace them with Any2Caption. In the absence of clear evidence of superior generalization, adaptability, or usability, the practical contribution and real-world impact of this work appear limited.

**Unreliable Evaluation**

The evaluation pipeline is also not convincing to me.

For text/caption quality, Table 3 already shows that higher caption metrics do not correlate with better video results, undermining the relevance of Table 4.

Although the appendix includes VDC benchmark results for captioning evaluation, the compared baselines are largely outdated, and notably, AuroraCap (introduced in the VDC paper itself and reported results better than Any2Caption) is missing. Thus VDC results is still not meaningful to me.

For video quality, most of the reported gains are minor and may fall within metric variance (many of these metrics are known to be not stable, e.g. aesthetic quality, etc), while qualitative examples in the paper are relatively limited considering the large amount of tasks the method claimed to tackle.

I also checked the provided videos in the supplementary materials, but in many cases I can't find significant differences/improvements comparing the short caption version and the structured caption version (e.g, camera to video, ids + depth to video).

Given these issues, the evaluation is relatively weak and potentially misleading. A proper human study (e.g. voting) comparing videos from short prompts, existing enhancers, and Any2Caption-generated captions would provide much more credible evidence, and a more qualitative comparison is also needed per task.

**Minor Issues**
- The camera pose visualization uses overly large frustum cones, making motion changes almost invisible.

- Several typos:
    - Fig 1 “normal bae” → “normal base”
    - Table 4 “METER” → “METEOR”
    - L307 “access” → “assess”?
    - Table 2 “vieo” → “video”

**Questions:**

- How is CLIP-T for long structured caption calculated (As CLIP textual encoder can not encode long sequence without losing information)? Is it still short-caption-video similarity score?

---

### Official Review · Reviewer_5Cq1 · 2025-10-30

**Soundness:** 3
**Presentation:** 3
**Contribution:** 3
**Rating:** 6
**Confidence:** 3

**Summary:**

The paper introduces Any2Caption, a novel framework designed to enhance controllable video generation by accurately interpreting diverse user inputs—such as text, images, videos, and specialized conditions like motion and camera poses—into dense, structured captions. The core idea is to decouple the interpretation of these multimodal conditions from the video synthesis process, leveraging modern multimodal large language models (MLLMs) to bridge the gap between user intent and video generation.

The authors also present Any2CapIns, a large-scale dataset containing 337K instances and 407K conditions, specifically curated for training Any2Caption in an any-condition-to-caption instruction tuning paradigm. Comprehensive evaluations demonstrate that Any2Caption significantly improves both the controllability and quality of generated videos compared to existing models. The framework can seamlessly integrate with various backend video generators without requiring additional fine-tuning, making it a versatile and efficient solution for controllable video generation.

**Strengths:**

1. This work presents a compelling and original formulation of controllable video generation by decoupling multimodal condition understanding from the video synthesis process, a design that leverages the strengths of modern MLLMs and avoids overburdening diffusion/DiT models with multimodal reasoning. This “any-condition-to-caption” paradigm is conceptually clean and practical, and represents a creative generalization of prompt enrichment and recaptioning techniques: rather than fine-tuning generators or relying solely on textual prompts, the system converts arbitrary input signals (text, images, depth, pose, camera motion, multiple identities) into structured captions to serve as universal control signals.

2. The paper demonstrates high technical quality with substantial engineering and system design effort, including the construction of Any2CapIns, a large multimodal dataset (337K videos, 407K condition annotations) curated through a mixture of GPT-4V and human verification. The method includes carefully considered architectural choices (dedicated pose and camera encoders, progressive mixed training, alignment stage) and ablations support these decisions. Empirical results are broad and convincing: the approach consistently improves controllability, instruction fidelity, and video quality across diverse backbones (e.g., HunyuanVideo, CogVideoX, Ctrl-Adapter, CameraCtrl) and generalizes to unseen controls such as sketches and segmentation masks.

3. The pipeline is well illustrated, training stages are clearly decomposed, and evaluation includes a mix of lexical, semantic, intent-based, and perceptual metrics, along with qualitative examples. The significance is high as controllable video generation is a rapidly developing area, and this architecture offers a plug-and-play solution that aligns with the trend toward modular generative stacks. The work has clear potential to influence future research, especially where multimodal control and human-intent grounding intersect.

**Weaknesses:**

1. A primary concern is dependence on MLLMs for accurate interpretation and structured captioning. While the paper shows strong results, caption hallucinations or subtle misinterpretations may degrade downstream generation; however, the paper does not deeply quantify or analyze such failure modes (e.g., how errors propagate through the pipeline). Providing a systematic robustness study—e.g., noisy or ambiguous conditions, conflicting signals between modalities—would strengthen confidence in real-world deployment.

2. While the plug-and-play nature is a strength, practical system overhead—running a large MLLM per request before video synthesis—is not quantified. Reporting latency costs or FLOPs would provide transparency regarding scalability for interactive creative workflows.

**Questions:**

1. Did the authors test dynamic or learned templates?

2. In scenarios where visual conditions convey information that is hard to verbalize (fine geometry, fashion textures), does text bottleneck expressiveness?

3. Have the authors conducted or considered user studies to verify usability and perceived control quality?

---

### Official Review · Reviewer_vmYy · 2025-10-31

**Soundness:** 2
**Presentation:** 3
**Contribution:** 3
**Rating:** 6
**Confidence:** 3

**Summary:**

In text + any condition to video generation, the alignment between text prompts and additional conditions is often overlooked. This paper proposes an MLLM fine-tuning recipe and dataset for generating structured text prompts that are aligned with additional conditions in video generation models.

**Strengths:**

- It addresses the alignment between text and additional non-text conditions, resulting in superior performance across various conditional tasks such as camera controllable video generation and depth-to-video generation.
- Extensive analysis and experiments are provided.
- The effects of structured captions and experimental strategies are well ablated.

**Weaknesses:**

Comparisons in video generation tasks are mostly limited to short naive captions (baseline) vs. proposed structured enriched captions. There is a concern that if structured captions generated from non-fine-tuned VLMs, or simply enriched (but not structured) prompts also achieve good performance on video generation tasks, the justification for the proposed method could be somewhat diminished. It would be helpful if the thorough comparisons with those possible approaches.

**Questions:**

What are the failure cases of the proposed method?

---

### Note · Authors · 2025-11-13

I have read and agree with the venue's withdrawal policy on behalf of myself and my co-authors.